# Enhanced performance of in-plane transition metal dichalcogenides monolayers by configuring local atomic structures

Yao Zhou [1,8], Jing Zhang [2,8], Erhong Song[3,8], Junhao Lin[4,8], Jiadong Zhou[5], Kazu Suenaga [6], Wu Zhou[7], Zheng Liu [5], Jianjun Liu [3✉], Jun Lou[2✉] & Hong Jin Fan [1✉]

The intrinsic activity of in-plane chalcogen atoms plays a significant role in the catalytic performance of transition metal dichalcogenides (TMDs). A rational modulation of the local configurations is essential to activating the in-plane chalcogen atoms but restricted by the high energy barrier to break the in-plane TM-X (X = chalcogen) bonds. Here, we theoretically design and experimentally realize the tuning of local configurations. The electron transfer capacity of local configurations is used to screen suitable TMDs materials for hydrogen evolution reaction (HER). Among various configurations, the triangular-shape cobalt atom cluster with a central sulfur vacancy ($3Co_{Mo}$-$V_S$) renders the distinct electrocatalytic performance of $MoS_2$ with much reduced overpotential and Tafel slope. The present study sheds light on deeper understanding of atomic-scale local configuration in TMDs and a methodology to boost the intrinsic activity of chalcogen atoms.

[1] School of Physical and Mathematical Sciences, Nanyang Technological University, Singapore 637371, Singapore. [2] Department of Materials Science and Nano Engineering, Rice University, Houston, TX 77005, USA. [3] Shanghai Institute of Ceramics, Chinese Academy of Sciences, 200050 Shanghai, China. [4] Shenzhen Key Laboratory of Advanced Quantum Functional Materials and Devices, Department of Physics, Southern University of Science and Technology, 518055 Shenzhen, China. [5] School of Materials Science and Engineering, Nanyang Technological University, Singapore 639798, Singapore. [6] National Institute of Advanced Industrial Science and Technology (AIST), Tsukuba 305-8565, Japan. [7] School of Physical Sciences and CAS Key Laboratory of Vacuum Sciences, University of Chinese Academy of Sciences, 100049 Beijing, China. [8]These authors contributed equally: Yao Zhou, Jing Zhang, Erhong Song, Junhao Lin. ✉email: jliu@mail.sic.ac.cn; jlou@rice.edu; fanhj@ntu.edu.sg

MoS₂ is a promising candidate to replace Pt for electro-catalytic hydrogen evolution reaction (HER) due to its environmental friendly and low cost characteristics[1–3]. While increasing the conductivity via forming heterojunction bi-layer with a conductive substrate can promote the overall catalytic performance[4–6], the performance of pristine MoS₂ is restricted by the density of active sites[7–10]. The pursuit to maximize MoS₂ utility inspires researchers to explore various ways to rouse the activity of inert sulfurs in the MoS₂ basal plane. For example, edge-site engineering[9,11], phase transformation[12,13], amorphiza-tion[14,15], and in-plane doping/vacancy modifications[16–20] have been reported. Notably, changing the local configurations[18,21–24] by introducing atomic defects (doping or vacancy) is preferable as defected MoS₂ exhibits better stability compared to transformed 1T′ phase[25] and amorphous MoS₂[10]. However, the reported activity of in-plane sulfur enhanced by local configuration mod-ification is still far from that of Pt-based catalysts[26]. This is because the large energy are required to break the in-plane Mo-S bonds. In fact, few types of atomic local configurations have been realized in the basal plane of MoS₂[27–30], so the tuning ability of local configurations is quite limited so far[16]. Hence, in order to improve the intrinsic activity of in-plane sulfur atoms, it is essential to understand the intrinsic correlation and explore new methodologies to enrich stable and highly efficient local configurations.

Herein, we conducts both computational and experimental investigations in order to establish a correlation between local configuration and the electrocatalytic activity of monolayer MoS₂. A group of stable local configurations with non-noble period-IV single atom or clusters (Co, Fe, V, and Cr) accompanying addi-tional sulfur vacancy in the in-plane domain of MoS₂ have been attained. Given the correlation between binding strength and local configurations electronegativity, the activity of in-plane sulfur can be regulated by electron transfer capacity of local configurations. The peculiar triangular-shape Co atom cluster surrounding one sulfur vacancy configuration (viz., $3Co_{Mo}-V_S$) is identified by both calculation and experiments to be most efficient to activate the inert sulfur sites. Correspondingly, a distinct enhancement in HER activity is achieved ($\eta_{10}$: 75 mV and Tafel: 57 mV dec$^{-1}$), exhibiting the highest intrinsic HER activity among MoS₂ mate-rials. The microcell HER measurements show a volcano-like relationship between content of specific local configuration and activity, which corroborates the optimized concentration of $3Co_{Mo}-V_S$. Therefore, as demonstrated in the present work, it is possible to further activate the in-plane sulfur sites by rational engineering of the local configurations. This result may provide a route to unleash the electrocatalytic potential of TMD materials for efficient hydrogen generation in acidic solutions.

## Results
**Design efficient and stable local configurations.** The activation of the basal plane in TMDs have been extensively studied to achieve the stable structure and enhance their catalytic activity[31–33]. Sulfur vacancy ($Vs$) on the surface is an electron donor and can induce a localized gap state in MoS₂. Below a critical carrier density, the transport of donor states is governed by nearest-neighbor hop-ping at high temperatures and variable-range hopping (VRH) at low temperatures[23,34–36]. Regional charge states around a defect structure are suggested to make an important contribution to regulating the catalytic activity. Based on the above analysis, we believe that it is reasonable to monitor the defects induced Bader charge fluctuation, and the H adsorption to define the active sites through DFT computational screening.

To study the TM and synergistic effect of $Vs$ on sulfur sites, we have considered six configurations including TM atoms (TM: Co,

V, Fe, and Cr; TM amounts from 1 to 3) with or without $Vs$ (structures see in Supplementary Fig. 1 and Supplementary Note 1) that are set as models to screen stable catalytic structures through DFT calculations. The hydrogen adsorption free energy ($\Delta G_H$) is an effective descriptor to predict the activity for various catalyst systems[37]. The ideal value of $\Delta G_H$ is 0 eV, which corresponds to a thermoneutral state of the adsorbed atomic hydrogen and efficient proton/electron transfer and hydrogen release[1]. The correspondingly calculated $\Delta G_H$ of monolayer MoS₂ with varied local configurations are further exhibited in Supplementary Fig. 2 and Supplementary Table 1, indicating the stronger H* adsorption on atomically structured MoS₂ than on intact MoS₂. In addition to intrinsic activity ($\Delta G_H$), structural stability affecting the final electrochemical durability of catalysts should be considered. Based on the formation energies of all possible configurations (Supplementary Fig. 3), the $3Co_{Mo}-Vs$, $3Fe_{Mo}-Vs$, $1V_{Mo}$ and $1Cr_{Mo}$ are identified as the most stable structures in the different possible TM-introduced MoS₂ (Fig. 1a). Together with the activity (value of $\Delta G_H$), the $3Co_{Mo}-Vs$ is expected to be the potential structure with both good stability and high activity. The predicted HER activity of MoS₂ with different local configurations following the trend $3Co_{Mo}-Vs > 1V_{Mo} > 3Fe_{Mo}-Vs > 1Cr_{Mo}$ (Fig. 1b). This trend remains the same with solvation correction, as demonstrated by our calculations with the implicit solvation model (Supplementary Fig. 4). The hydrogen adsorption free energy ($\Delta G_H$) on basal plane of intact MoS₂ is far away from the optimal value. After tuning by local configurations, the $\Delta G_H$ value of −0.085 eV comparable to that of Pt[38], is achieved due to the much stronger bonding strength in S atoms with the assistance of $3Co_{Mo}-Vs$, which surpasses predicted activity of edge sites[8]. As expected, different configurations induce varied activity; all the structures with co-existence of $Vs$ and $TM_{Mo}$ atoms synergistically tunes the $\Delta G_H$ when compared to single one (Supplementary Table 1). In addition, the monolayers are used instead of porous 3D materials, minimizing the double layer effect induced by porosity. It is supported by CV curves in non-Faradaic region (blue rectangular in Supplemen-tary Fig. 5) with nearly no hysteresis loop.

It is important to reveal the underlying mechanism of enhanced catalytic activity due to the local configuration. The above analysis indicates that the defects (TM substitution and S-vacancy) and H adsorption could induce charge fluctuation of the regional structure due to electron delocalization of MoS₂. In principle, the catalytic activity depends on the charge transfer capacity before and after H adsorption. To identify the effective catalytic structure, we show the nearest and the next-neighbor atoms which possibly induce a charge fluctuation in HER (Supplementary Fig. 6 and Supplementary Note 2). First, the nearest metals (nMo and doped $(3–n)$TM, $n = 0$, 1, 2) and the adsorption S1 atom have relatively large change in charge (Fig. 1d–f, Supplementary Figs. 7–10). As a result, we consider $(3–n)$TM–S–nMo as the first-order catalytic structure that is comprised of TM substitutes, adsorption S atom, and the nearest Mo atoms. In contrat, the change in charge for the next-neighbor S and Mo atoms is relatively small. So they are considered as the second-order structure as the distance from the adsorption site is large. As a result, it is reasonable to assume the charge regulation of the second-order catalytic structure has a negligible effect on that of the first-order one. The radial distributions of charge distribution are presented in Supplementary Figs. 7–10. There-fore, in our study, we calculate the total charge difference of adsorption S atom and the nearest metals to depict the charge transfer capacity to S–H bonds. The amount of charge transfer of local configuration (namely, atoms to induce the charge transfer includes: nearest nMo, doped $(3–n)$TM, $n = 0$, 1, 2 and adsorption S1 atom) is linearly correlated with $\Delta G_H$. This result

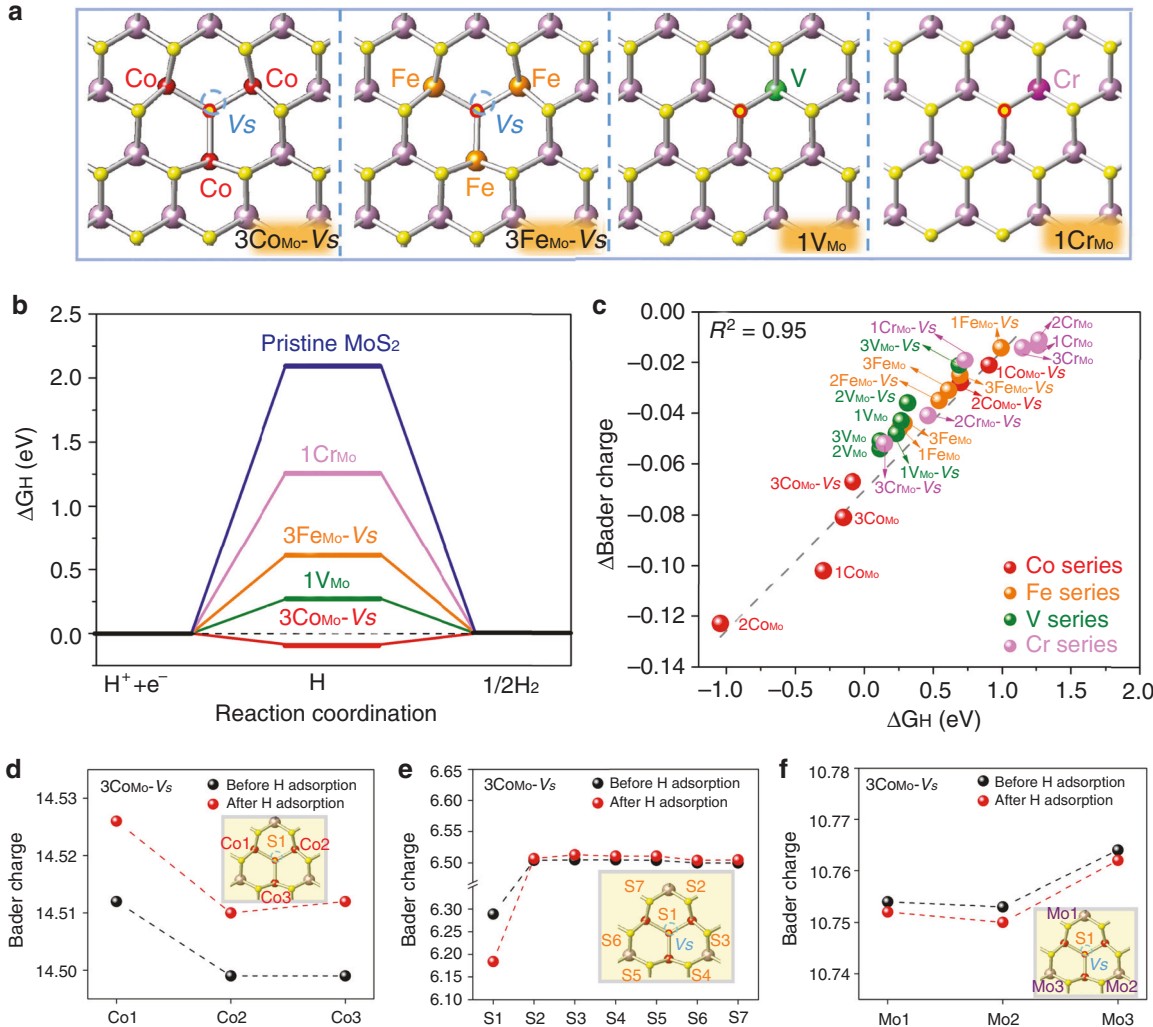

**Fig. 1 DFT calculations to predict effect of local configurations to HER activity of MoS₂. a** The most stable structures of MoS₂ with $3Co_{Mo}-Vs$, $3Fe_{Mo}-Vs$, $1V_{Mo}$, and $1Cr_{Mo}$ configurations and the S bonding with H is marked as red circles. **b** The free energy diagram of corresponding configurations and pristine MoS₂. **c** The correlation between change of Bader charge of local configuration around sulfur atoms and hydrogen adsorption free energy ($\Delta G_H$). The dashed line is linearly fitted with $R^2 = 0.95$. The Bader charge changes of **d** Co atoms, **e** S1 and next-neighbor S atoms, and **f** the next-neighbor Mo atoms when $3Co_{Mo}-Vs$ is introduced in the MoS₂.

indicates a charge regulation effect by the local configuration on HER activity (Fig. 1c). The linear correlation indicates that charge transfer capacity induced by varied local configurations are mainly delocalized in the first-order catalytic structure instead of on individual sulfur atoms. We found that, a charge difference around $0.07e^-$ (which corresponds to $\Delta G_H = 0$ eV) should correspond to a high HER catalytic activity.

**Realization and characterizations of local configurations.** In light of the superior activity induced by the predicted local configurations, we employ the chemical vapor deposition (CVD) method to synthesize several monolayer MoS₂ samples with various in-plane local configurations (Methods). The optical images of Co, Fe, Cr, and V-containing MoS₂ monolayers are shown in Supplementary Fig. 11a–d. Raman spectra confirm that all the as-prepared samples preserve the lattice structure of MoS₂ (Supplementary Fig. 12), as seen from the characteristic $A_{1g}$ mode at ~401 cm⁻¹ and the $E_{2g}^1$ mode at ~381 cm⁻¹ observed in pristine MoS₂ monolayer[39]. In addition, the Raman mappings indicate homogenous elemental distribution (Supplementary

Fig. 13). Atomic force microscopy (AFM) measurements further confirm that the as-prepared MoS₂ domains are monolayers (Fig. 2a–d) with a thickness ranging between 0.7 and 0.9 nm. As for the the system with small doping concentration, the peak-shift is ascribed to the dopant induced Fermi level movement[40]. However, the shifts of X-ray photoelectron spectra in both Mo 3d and S 2p are very small (below 0.3 eV, see Supplementary Fig. 14), likely due to the low dopant concentrations. Therefore the minor peak shift cannot justify if the dopants incorporate into the MoS₂ lattice. More evidence is provided by the high-resolution spectra of TM 2p (Supplementary Fig. 15), which show clearly the formation of metal-sulfur bonds in all samples and support the substitutional dopants within the MoS₂ lattice.

The annular dark-field (ADF) scanning transmission electron microscopy (STEM) imaging and electron energy loss spectroscopy (EELS) are used to further confirm the local atomic configurations. Figure 2e–h show the atomic structure of the Co-, Fe-, V-, and Cr-containing MoS₂ monolayers, respectively. All four images show the lattice of MoS₂ with Mo and S atoms alternating in bright and dim spots periodically. The TM atoms, which occupy the metal sites, show lower image contrast than

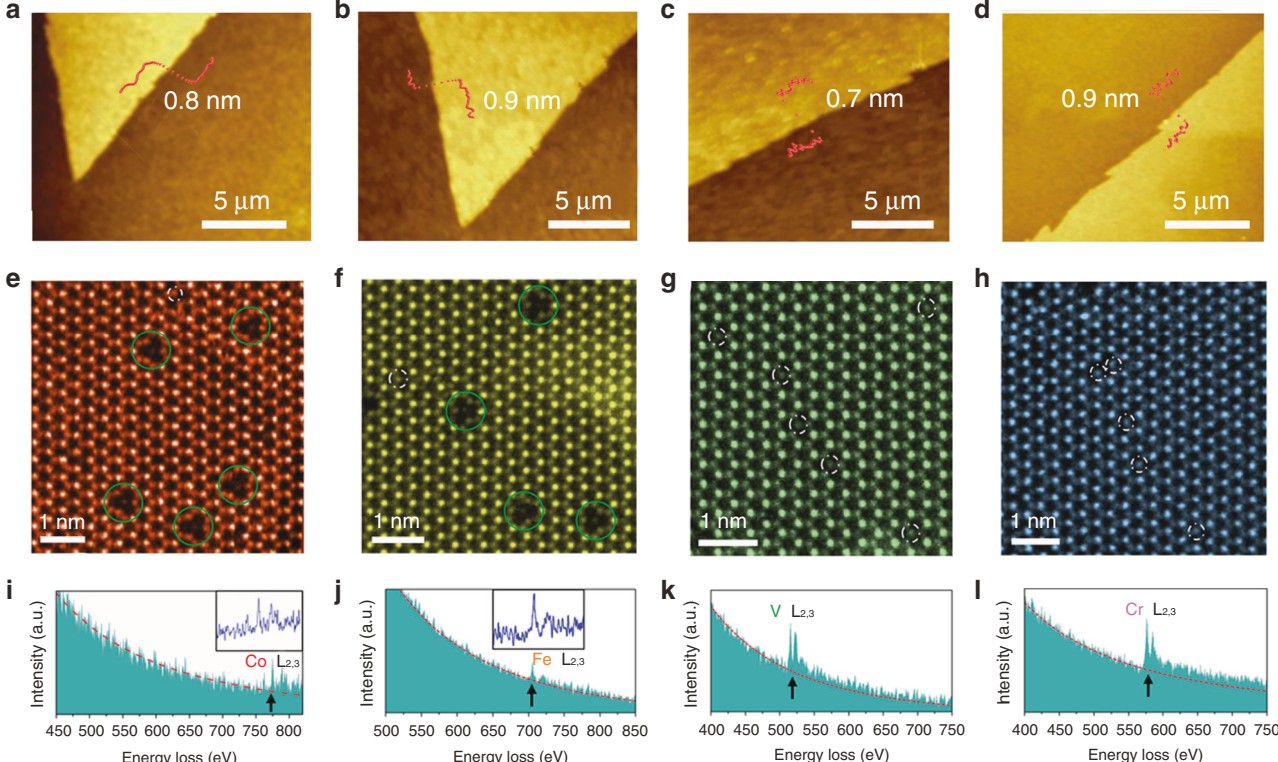

**Fig. 2 Imaging of MoS$_2$ monolayers with various local configurations.** AFM images of **a** 3Co$_{Mo}$−$Vs$, **b** 3Fe$_{Mo}$−$Vs$, **c** 1V$_{Mo}$, and **d** 1Cr$_{Mo}$ samples, illustrating the monolayer nature of the as-synthesized MoS$_2$. **e–h** Atomic resolution STEM images of the TM-containing MoS$_2$ samples with **i–l** corresponding electron energy loss spectrum on single TM. The images confirm that, while Co and Fe atoms prefer to form triangular clusters where three TM atoms connect a central sulfur vacancy (3TM$_{Mo}$−$Vs$, highlighted by green circles), V and Cr only form substitutional single atom without sulfur vacancy (1TM$_{Mo}$, highlighted by white dashed circles), in good consistence with the theoretical prediction.

typical Mo atoms but similar to S$_2$ columns due to the nature of the STEM imaging[27]. The line intensity profile of the single isolated dopant site and comparison to the STEM simulation (Supplementary Fig. 16) confirm that the TM atoms successfully doped into the lattice rather than the adatoms on the surface of MoS$_2$. A more careful inspection reveals two main types of local configurations as predicted by previous theoretical calculations: one TM atom substitutes the Mo site forming an isolated single TM site, as marked by the white circles in all four TM-containing MoS$_2$ monolayers (Fig. 2g, h, k, l); three TM atoms forming a triangular cluster with a connecting central sulfur vacancy, named as 3TM$_{Mo}$−$Vs$, as highlighted by green circles in Co and Fe-containing system (Fig. 2e, f, i, j). However, we find the 3TM$_{Mo}$−$Vs$ are the dominating configuration in the Co- and Fe-containing MoS$_2$ monolayers. The single isolated TM sites account for very small ratio (<10%) compared to the 3TM$_{Mo}$−$Vs$ (Supplementary Figs. 17, 20c, and 21c; Supplementary Notes 3, 6, and 7), which corroborates that the latter are the dominating causes towards the HER activity.

On the other hand, Cr and V form predominantly isolated single TM sites in the MoS$_2$ lattice. This is due to the different formation energy of the two types of local configurations with different TM atoms. The single atom EELS measurements on the TM in each image further confirm the chemical identity of the corresponding introduced element, as recognized by the sharp L edges of Co, Fe, V, and Cr, respectively, offering strong evidence of the presence of TM atoms and the consistence of the predicted local configurations. The reference EELS spectra taken away from the dopant site (Supplementary Fig. 18 and Supplementary Note 4) confirms the observed sharp peaks in the spectra of Fig. 2 are not an artifact during the collection at the dopant site.

**Electrochemical test of MoS$_2$ with local configurations**. To verify the predicted HER activity of in-plane sulfur modulated by designed local configurations, the effects of 3Co$_{Mo}$−$Vs$, 3Fe$_{Mo}$−$Vs$, 1Cr$_{Mo}$, and 1V$_{Mo}$ on HER catalytic activity are examined using a three-electrode electrochemical cell in an electrolyte containing 0.5 M H$_2$SO$_4$. Data are compared to pristine MoS$_2$ and commercial Pt/C. Before LSV tests, electrochemical activation was implemented. Stable CVs of configured MoS$_2$ samples after electrochemical activation process (black curves in Supplementary Fig. 19 and Supplementary Note 5) indicate no phase change during activation. Figure 3a shows linear-sweep voltammograms (LSV) in the cathodic direction after the correction of ohmic potential drop (i.e., iR), where the currents are normalized to the electrode geometric area. It is seen that the pristine MoS$_2$ with an overpotential of 317 mV at 10 mA cm$^{-2}$ ($\eta_{10}$) shows an inferior HER activity than those of configured MoS$_2$. The optical image of pristine MoS$_2$ as shown in Supplementary Fig. 22 shows the similar edge length with that of configured MoS$_2$, excluding the edge effect on different activity. Both the 3Fe$_{Mo}$−$Vs$ and the 1Cr$_{Mo}$ samples exhibit values of $\eta_{10}$ over 200 mV. On the contrary, the 3Co$_{Mo}$−$Vs$ and the 1V$_{Mo}$ show significantly reduced $\eta_{10}$ values down to below 150 mV. In particular, the 3Co$_{Mo}$−$Vs$ has a lowest $\eta_{10}$ value of only 75 mV. The first cycles of CV for HER were operated to verify the structural stability of introduced basal configurations after activation (Supplementary Fig. 19). The trend of activity shown in the first CVs of configured MoS$_2$ exhibits the same as that of LSV curves. Note that the Co atoms in 3Co$_{Mo}$−$Vs$ configuration has an over 90% occupancy among the total Co atoms. Hence, even though a minor content (<0.2 at%) of 1Co configuration is detected, we believe the 3Co$_{Mo}$−$Vs$ configuration plays the

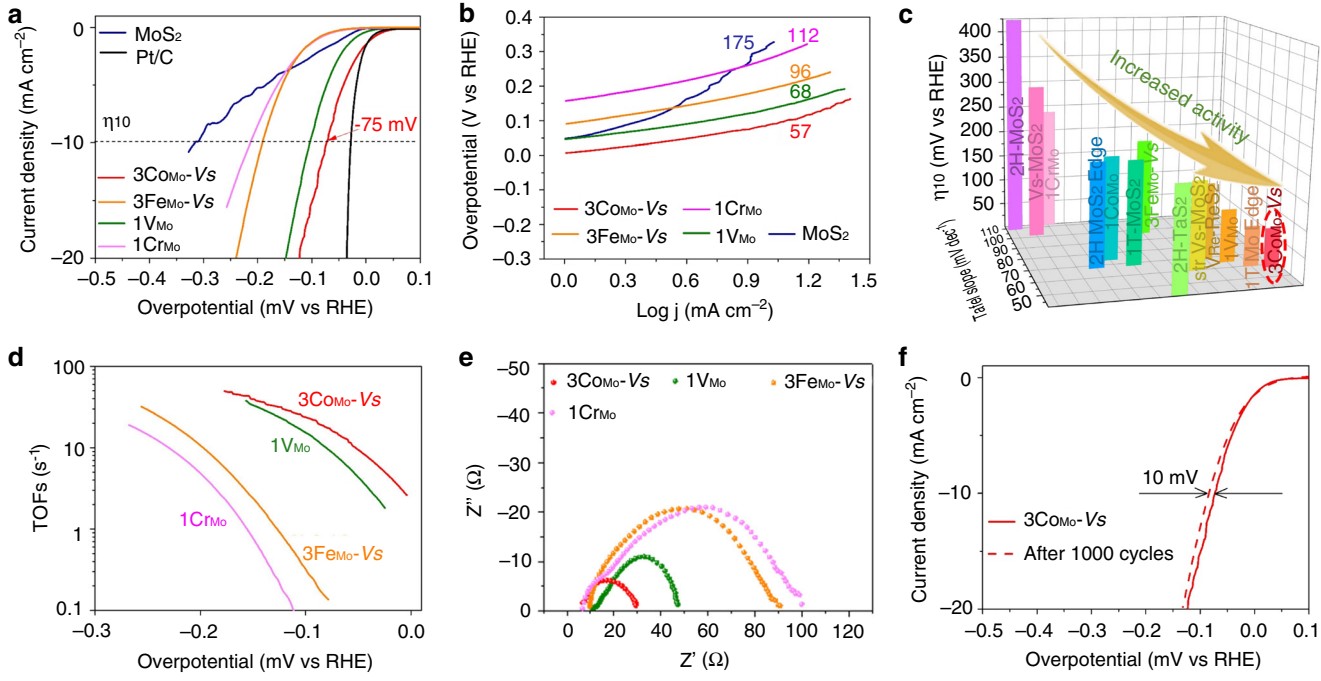

**Fig. 3 HER properties of samples with different local configuration types. a** Polarization curves of pristine $MoS_2$, $MoS_2$ with $3Co_{Mo}-Vs$, $3Fe_{Mo}-Vs$, $1V_{Mo}$, $1Cr_{Mo}$ configurations and Pt/C. The currents are normalized to the projected geometric area of the electrode. **b** The corresponding Tafel plots of the polarization curves in **a**. **c** Comparison of $\eta_{10}$-Tafel slope for HER catalysts in 0.5 M $H_2SO_4$. $MoS_2$ with $3Co_{Mo}-Vs$ configuration exhibits top performance. The data are taken from refs. [16,42–48]. **d** TOFs of $MoS_2$ with varied configuration types. **e** Electrochemical impedance spectroscopy (EIS) Nyquist plots for sample $3Co_{Mo}-Vs$, $3Fe_{Mo}-Vs$, $1V_{Mo}$, and $1Cr_{Mo}$. **f** Long-test stability test for the $3Co_{Mo}-Vs$ electrode at a current density of 10 mA cm$^{-2}$. .

dominating role in the catalyst activity. This is also in concise with the calculation that $\Delta G_H$ of $3Co_{Mo}-Vs$ is closest to 0 eV (Supplementary Table 1). The corresponding Tafel plots show the same trend with that of $\eta_{10}$ (Fig. 3b). The $3Co_{Mo}-Vs$ configuration sharply reduces the Tafel slope from 175 mV dec$^{-1}$ in pristine $MoS_2$ to 57 mV dec$^{-1}$. And the $1V_{Mo}$ sample gives an acceptable value of 68 mV dec$^{-1}$. In comparison, the $3Fe_{Mo}-Vs$ and $1Cr_{Mo}$ configurations have little effect to the Tafel slope. Therefore, the lowered Tafel slopes of the $3Co_{Mo}-Vs$ and $1V_{Mo}$ with a fast discharge process of protons[41] (Supplementary Note 8), may reflect a strengthened capability to adsorb H. The Faradic efficiency was determined from the produced $H_2$ characterized quantitatively by gas chromatography. As shown in Supplementary Fig. 23, the $3Co_{Mo}-Vs$ sample exhibits >98% efficiency over the time scale of the measurement, confirming the $H_2$ as the dominating product during the whole electrolysis process. From the overall comparison (Fig. 3c), we can conclude that the $3Co_{Mo}-Vs$ configuration, with synergistic triangular Co clusters surrounding one $Vs$ in the center, renders $MoS_2$ monolayer the best HER catalytic performance among all the configured $MoS_2$ samples[16,42–48]. Supplementary Table 2 provides an extensive comparison to other TMDs and non-noble metal catalysts in their electrocatalysis of HER. The performance of our $MoS_2$ monolayer with $3Co_{Mo}-Vs$ configuration exceeds all the pure TMD monolayer catalysts, and also compete with other non-noble metal catalysts.

The turnover frequency (TOF) per sulfur is calculated in order to correlate the intrinsic activity per sulfur atom with the local configuration (Supplementary Note 9). Each sulfur site tuned by $3Co_{Mo}-Vs$ or $1V_{Mo}$ possesses much higher efficiency than that in $3Fe_{Mo}-Vs$ and $1Cr_{Mo}$ samples with increased value of TOFs. Compared to other configurations, the $3Co_{Mo}-Vs$ and $1V_{Mo}$ samples demonstrate the most appropriate tuning on the charge transfer capacity of local configuration.

Electrochemical impedance spectroscopy with fitted circuit models (Supplementary Fig. 24) shows significantly decreased charge-transfer resistances ($R_{ct}$) for the $3Co_{Mo}-Vs$ (30.3 $\Omega$) and the $1V_{Mo}$ (47.9 $\Omega$) samples, as compared to those of $3Fe_{Mo}-Vs$ (91.5 $\Omega$) and $1Cr_{Mo}$ (100.3 $\Omega$), indicating a facilitated charge transfer between the S and protons in electrolyte (Fig. 3e). In addition, the $3Co_{Mo}-Vs$ sample exhibits an extraordinary long-term operation durability with small changes in potential (Fig. 3f and Supplementary Fig. 25). Hence, we may conclude that the $3Co_{Mo}-Vs$ configuration is efficient for HER during the whole cycling process.

**Microcell measurements.** In addition to the effect of local configuration type, it is expected that the concentration of such local configurations can also influence the amount of active sulfur sites. In order to prove the concentration effect, the on-chip electrochemical micro-devices are fabricated from a set of $3Co_{Mo}-Vs$ samples with local probe test, as shown in Fig. 4. Figure 4a, b show the three-electrode setup for the electrochemical measurements (more details are shown in Supplementary Fig. 26). In the first step, a controlled linear $I–V$ scan is done on the PMMA layer (Supplementary Fig. 27) to measure the electrochemical blocking reliability of the PMMA layer. The $MoS_2$ samples with different $3Co_{Mo}-Vs$ concentrations (determined from STEM measurements, see Supplementary Fig. 21 and Supplementary Table 3) are applied for the microcell experiments. The obtained results are Fig. 4c–e. We can clearly see that an optimal Co concentration, corresponding to $\eta_{10} < 100$ mV, should be around 3.8 at%, which translates to the $3Co_{Mo}-Vs$ concentration of ~1.2 at%. The excessive increase in $3Co_{Mo}-Vs$ concentration results in performance decline. Possible reasons for this are deteriorated surface stability[16] and superfluous lattice distortion[18]. This concentration effect also corroborates the key contribution of $3Co_{Mo}-Vs$ rather

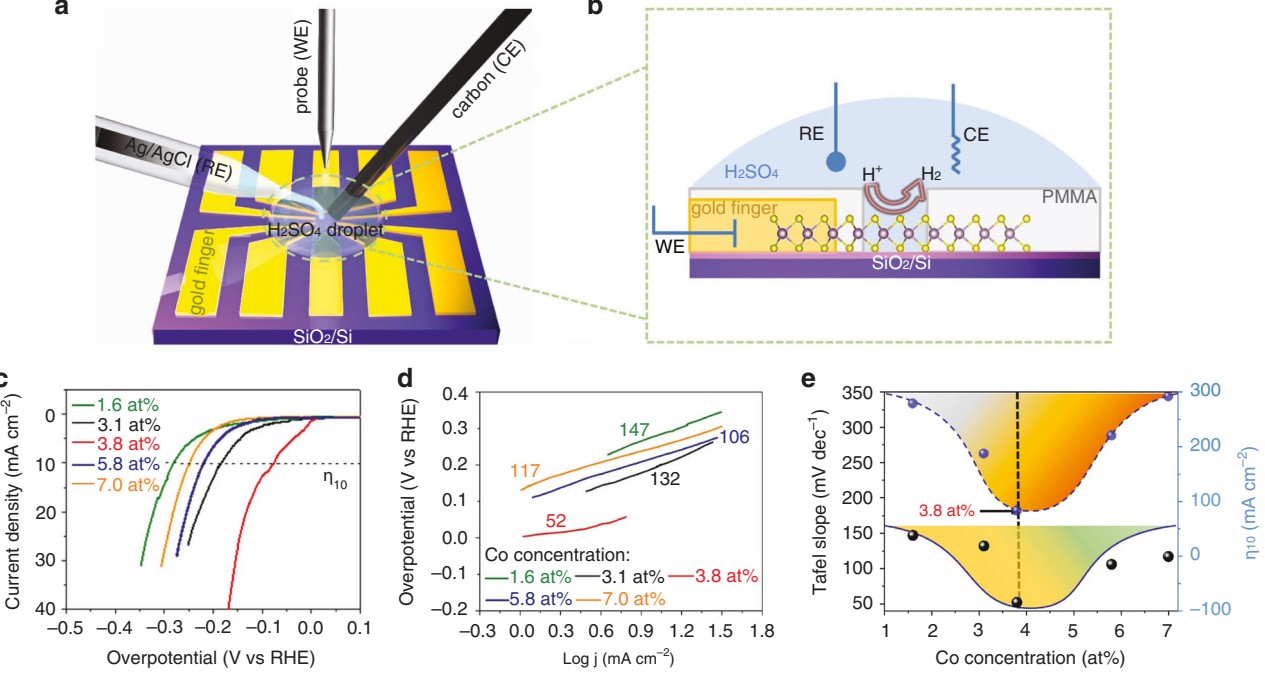

**Fig. 4 Microcell HER measurements of the $3Co_{Mo}-Vs$ samples. a, b** Schematic illustration of the microreactor device fabricated on $3Co_{Mo}-Vs$ structured $MoS_2$ monolayer. **c** LSV curves of the samples with Co concentrations of 1.6, 3.1, 3.8, 5.8, and 7.0 at%. **d** The corresponding Tafel plots of the polarization curves in panel **c**. **e** The changing trends of Tafel slopes and $\eta_{10}$ with varied Co concentration. The optimal Co concentration is determined to be ~3.8 at%, corresponding to the concentration of $3Co_{Mo}-Vs$ around 1.2 at%.

than $1Co_{Mo}$ to the HER catalyst activity enhancement. In addition, the microcell HER measurements are implemented on $3Co_{Mo}-Vs$, $3Fe_{Mo}-Vs$, $1V_{Mo}$, and $1Cr_{Mo}$ samples with similar defect concentrations (Supplementary Fig. 28), showing the similar trend with that of three-electrode measurements (Fig. 3a).

## Discussion

We have promoted the per-site electrochemical activity of in-plane sulfur sites of $MoS_2$ monolayer via tuning H–S bonding strength, which can be understood by a hypothetical model of activating the inert sulfur atom into an open valence state. That activation can be correlated with the charge transfer capacity of local configuration. This is realized by forming various local configurations of transition metal atom or clusters (Co, Fe, V, and Cr) and compensative sulfur vacancy ($Vs$), which are confirmed by STEM images. In particular, the in-plane sulfur atoms modulated by $3Co_{Mo}-Vs$ configuration render the most active $MoS_2$-based HER electrocatalyst in acidic medium to date (an overpotential $\eta_{10}$ of 75 mV). The optimized $3Co_{Mo}-Vs$ configuration is also verified by the systematic DFT calculations. In addition, the suitable Co concentration for the HER performance is achieved by the in-situ probe measurements of microcell. Our work highlights the potency of local configuration engineering in boosting the in-plane electrocatalytic activity of $MoS_2$, as well as possibly other 2D TMD monolayers.

## Methods

**Synthesis of configured and pristine $MoS_2$.** Pure $MoS_2$ and configured $MoS_2$ were synthesized by CVD method using $MoO_3$ and sulfur (Sigma) as the precursor. For different TM-doped $MoS_2$, $V_2O_5$, $CrCl_3$, $Fe_2O_3$, and $Co_3O_4$ were used as the corresponding TM sources. The synthesis was conducted using a quartz-tube single-zone furnace (1-inch diameter) in a temperature range from 550 to 650 °C. Specifically, for the growth of pure $MoS_2$, a quartz boat containing 10 mg $MoO_3$ powder was put in the center of the tube, and the $SiO_2/Si$ substrate was placed on top of the quartz boat with the front side facing down. Another quartz boat containing 0.5 g sulfur powder was put upstream. The temperature ramped up to 700 °C in 15 min, and was maintained at the peak temperature for 5 min to 10 min.

During the reaction, a constant 80 sccm Ar flow was used as the carrier gas. After the reaction, the furnace cooled down naturally to room temperature. For the $3Co_{Mo}-Vs$, $3Fe_{Mo}-Vs$, $1V_{Mo}$, and $1Cr_{Mo}$ structured $MoS_2$, the precursor loaded in the central boat contained mixed powder of $V_2O_5$, $CrCl_3$, $Fe_2O_3$, $Co_3O_4$, respectively, with $MoO_3$ (mole ratio of 2: 98). The carrier gas used for the structured $MoS_2$ was mixed $Ar/H_2$ with a flow of 80/5 sccm. The rest reaction conditions were the same as that for pure $MoS_2$.

**Structural characterizations.** Room temperature Raman measurements were performed using a WITEC alpha 300 R Confocal Raman system with an excitation laser of 532 nm. The Raman system was pre-calibrated based on the Raman peak of crystalline Si at 520 cm$^{-1}$. The laser power was kept below 1 mW to avoid sample heating. The TEM samples of the $3Co_{Mo}-Vs$, $3Fe_{Mo}-Vs$, $1V_{Mo}$, and $1Cr_{Mo}$ structured $MoS_2$ were prepared as follows. A layer of poly (methyl methacrylate) (PMMA) was spin-coated on the sample surface with a thickness of ~1 μm, and then baked in an electric oven at 180 °C for 3 min. Afterwards, the substrates were immersed in a NaOH solution (1 M) overnight to dissolve the $SiO_2$ layer. After lift-off, the $MoS_2$ samples were washed with DI water for several cycles. Then the monolayer samples were fished by a TEM grid (Quantifoil Mo grid). The obtained TEM specimen were dried naturally in ambient environment, and then dipped into high-purity acetone overnight to remove the PMMA layers. The STEM investigation was performed at room temperature on an aberration-corrected Nion UltraSTEM-100 and a JEOL 2100 F with a cold field-emission gun and an aberration corrector (the DELTA-corrector), both operating at 60 kV.

**Electrochemical measurements.** PMMA methylbenzene was uniformly spun on the $SiO_2/Si$ substrates deposited with monolayer $MoS_2$. After baking at 100 °C for 5 min, the PMMA film covered substrates were immersed in a 5 M KOH solution. As a result of the etching effect by KOH, the monolayer $MoS_2$ samples with the PMMA film were detached from the $SiO_2/Si$ substrate. Then, the obtained monolayer $MoS_2$/PMMA films were washed in DI water and overlaid on the glassy carbon rotating disk electrode (RDE). After the thorough evaporation of DI water between the RDE electrode and the $MoS_2$/PMMA films, the PMMA films were further removed by dipping into acetone. As a result, the glassy carbon RDE electrode covered by monolayer $MoS_2$ were obtained[45,46].

For the electrochemical measurements, a standard three-electrode cell consisting of the glassy carbon RDE as the working electrode, a graphite carbon counter electrode and a saturated calomel reference electrode (SCE) was used. The electrolyte solution was 0.5 M $H_2SO_4$. An electrochemical workstation (CHI760) coupled with a RDE system (AFMSRCE3529, Pine Research Instrumentation, USA) was used to control the cell. The potential versus the reversible hydrogen electrode (RHE) was calculated according to $E_{RHE} = E_{SCE} + E°_{SCE}$ (0.2412) + 0.059 × pH. Before HER test, the catalysts went through an electrochemical

activation process by cyclic voltammetry scanning in the same electrolyte (0.5 M $H_2SO_4$) with a scan rate of 100 mV s$^{-1}$ in the potential range of 0.1 to $-0.29$ V (vs. RHE). Linear sweep voltammetry (LSV) measurements were conducted with a scan rate of 2 mV s$^{-1}$ under 1500 rpm. The current vs. potential plots were corrected by 90% ohmic compensation. The electrochemical impedance spectroscopy (EIS) were obtained in the same three-electrode configuration in the frequency range of 100 KHz to 0.1 Hz and at an applied current of 10 mA cm$^{-2}$. For the stability assessment, polarization data were measured in the beginning and after 1000 CV sweeps ($-0.2$ and $+0.2$ V vs. RHE, scan rate: 50 mV s$^{-1}$). In addition, the constant-current (10 mA cm$^{-2}$) measurements were also implemented to evaluate the stability of potential.

The on-chip electrochemical measurements were carried out following the previous report[11]. Briefly, structured-MoS$_2$ monolayers were transferred onto SiO$_2$ (300 nm)/Si substrate with pre-made gold fingers by PMMA assisted wet transfer method. Monolayers were further patterned into domains by e-beam lithography and 5-s treatment in nitrogen plasma. Contacts between gold fingers and monolayers were made via e-beam lithography and gold deposition processes. Microcell reaction windows were made by e-beam lithography on 1-μm-thick spin-coated PMMA layer. During measurements, gold fingers connecting configured MoS$_2$ monolayers, Ag/AgCl encapsulated by Luggin capillary and carbon rod were used as working, reference and counter electrodes respectively. In all, 5 μl of 0.5 M $H_2SO_4$ (degassed with Ar bubbling for 10 min) was used as electrolyte for each test. The scan rate for the LSV tests were 10 mV/s.

**Computational methods.** The density functional theory (DFT) calculations were perfomed using the Vienna Ab initio simulation package[49,50]. The generalized gradient approximation with the Perdew−Burke−Ernzerhof exchange−correlation fuctional and a 450-eV cutoff for the plane-wave basis set are employed[51]. The projector-augmented plane wave was adopted to describe the electron−ion interactions[52]. The em piracal dispersions of Grimme (DFT-D2) was applied to account for the long-range van der Waals interacions[53]. All calculations were spin-polarized and the convergence threshold was set to be 10$^{-4}$ eV in energy and 0.01 eV/Å. The k-point sampling of the Brillouin zone was obtained using a 4 × 4 × 1 by the Monkhorst−Pack scheme. In addition, a 5 × 5 × 1 supercell was also used to confirm the sufficiency of 4 × 4 × 1 supercell (Supplementary Fig. 29). In the electronic structure calculation, denser k-points (8 × 8 × 1) were used for better accuracy. The vacuum slab of 15 Å was inserted in the z-direction for suface isolation to eliminate periodic interaction. The free energy of the adorbed state was calculated as

$$\Delta G = \Delta E_{H^*} + \Delta E_{ZPE} - T\Delta S, \tag{1}$$

where $\Delta E_{H^*}$ is the hydrogen chemisorption energy, and $\Delta E_{ZPE}$ is the difference of the zero point energy between the adsorbed state and the gas phase. Considering the fact the vibriation entropy of H* in the adsorbed state is very small, the entropy of 1/2 H$_2$ adsorption can be approximated as $\Delta S_H \approx -1/2 S_{H^2}^0$, where $S_{H^2}^0$ is the entropy of H$_2$ in the gas phase at the standard conditions.

## Data availability

All relevant data are available from the authors.

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

## Acknowledgements

H.J.F. and Y.Z. thank the financial support by Singapore MOE under its Tier 2 grant (MOE2017-T2-1-073). Y.Z. appreciates the financial support from the Shanghai Natural Science Foundation of China (19ZR1465100). E.H.S. and J.J.L. thank the financial support from National Natural Science Foundation (21973107, 51702345). J.D.Z. and Z.L. thank the financial support from MOE Tier 2 grant MOE2016-T2-1-131. J.H.L. thanks the support from National Natural Science Foundation of China (11974156) and Guandong International Science Collaboration Project (Grant No. 2019A050510001). W.Z. thanks the financial support from National Key R&D Program of China (2018YFA035800). K.S. acknowledges JST-ACCEL and JSPS KAKENHI (JP16H06333, JP25107003, and P16382) for financial support. J.Z. and J.L. appreciate the financial support from Welch Foundation C-1716 and the NSF I/UCRC Center for Atomically Thin Multifunctional Coatings (ATOMIC) under award # IIP-1539999. We appreciate the supports of X.G.Li and S.S. Tang from Nanyang Technological University on the measurements of Faradic efficiency.

## Author contributions

Y.Z., Z.L., J.L., and H.J.F. conceived the project. Y.Z., J.Z., and J.D.Z. designed the experiments. J.H. L.,W.Z., and K.S. performed the STEM characterization of the samples and data analysis. E.H.S. and J.J.L. proposed the local configuration electroneagtivity and carried out the theoretical calculations. Y.Z., J.Z., E.H.S. and J.J.L. wrote the paper with contributions from other co-authors. All authors discussed and commented on the manuscript.

## Competing interests

The authors declare no competing interests.
