## [Peer Review File · Nature Communications]

Reviewers' comments:

Reviewer #1 (Remarks to the Author):

This manuscript reports on atomic-engineering of defects on the basal plane of monolayer-MoS₂, for improving HER catalytic-efficiency. The manuscript articulates the What-Why-How of its research design, with a rich set of computational and experimental data. The data interpretations are, in my opinion, at least 90% sound. Nevertheless, I am reluctant to offer the authors my favorable recommendation for the publication of the manuscript in NComms, with the following reasons:

The pillar of this work is "The intrinsic per-site activity of in-plane chalcogen atoms dominate the hydrogen evolution reaction (HER) catalytic performance of transition metal dichalcogenides (TMDs)" {the first sentence in the Abstract}. With this pillar, the structure of the work on developing the triangular-shape cobalt atom cluster with a central sulfur vacancy in MoS₂ basal plane as the best HER-catalysis is erected. I opine that this pillar is by no means rigid. Certainly, in-plane defect-engineering constitutes one approach in enhancing the performance of TMDs. However, there are other options as important as this "in-plane-defect-engineering" approach, options such as the "edge-site-engineering" option, and the "heterojunction-bilayer" option. In fact, the authors' claim of getting the record-best Tafel slope of 57 mV/dec in their work is not valid because the "heterojunction-bilayer" option, with MoS₂/graphene, is known to give a Tafel slope of 41 mV/dec. In this context, the "heterojunction-bilayer" option should be considered as "dominating". These prior-arts have already been reviewed by Zhu et al. { Chem Soc Rev 47, 4332 (2018) }, an article which is also cited in this manuscript.

In practical HER, a catalyst for the reaction must be loaded on an electrode. The authors of this manuscript take the approach of making relative large TMD layers (a scale of many microns) and decorating the unreactive basal planes with "3Co-VS" for enhancing HER-performance. The preparation of such large flakes of TMDs typically requires CVD with adequate control of crystal growth, as shown in the manuscript. Transferring these flakes onto HER electrodes is not trivial. In comparison, many means including various solvothermal methods and CVD methods can yield nano-TMD-polygons (e.g., triangular clusters) which can be precipitated directly onto HER electrodes. When such nano-TMD-polygons merely have a few metal-atoms per edge, the number of HER-active-sites (edge-sites) per unit mass can become extremely high. In fact, a recent computational work has given quantitative details of the science and technology of this "edge-site-engineering" approach {An et al., Nano Lett. 17, 368(2017)}. In principle, the number of active sites per unit mass of TMD by synthesizing TMD-triangles (particularly for VS₂-triangles) with a size-scale not more than 2nm will be more than that by synthesizing micron-size TMD layers with around 2% of "3Co-VS".

Further, I would like to add several technical comments:

1. The scientific root of overpotential in TMD-electrochemistry is surface band-bending and location of Fermi level on the TMD-surface in the electrochemical-bath. The authors should correlate their DOS and band-structure results to overpotential-lowering, as overpotential-lowering is the value-proposition of this work.
2. The authors show the XPS spectra of Mo and claim that the spectra give peak-shifts are derived from the chemical presence of defects such as "3Co-VS". In reality, for analyses of semiconductors such as TMDs, XPS spectra all place their zero point of "Binding Energy" at Fermi level. Hence, the Mo XPS peak of a p-MoS₂ sample may have a Binding-Energy peak position more than 1eV lower than that of an n-MoS₂ sample. Since the "defect" concentration in this work is typically merely about 2% or so, spectral differences due to chemical-compositional changes from MoS₂ to MoS₂ having 2% defects can hardly be directly measured. The peak-shifts observed in this work must be due to the changes in Fermi level induced by defect-engineering. Indeed, the DOS computational data show that Fermi level moves towards CBM when "3Co-VS" is introduced to MoS₂. This Fermi level movement is shown in the XPS Mo spectra.

Reviewer #2 (Remarks to the Author):

Zhou et al. report the catalytic performance of monolayer MoS₂ with different local configuration of sulfur vacancies, including 3Co-V, 3Fe-V, 1V, 1 Cr, and pristine MoS₂. They first performed theoretical simulation and find out preferred local configuration, which is 3Co-V. Then they synthesized monolayer MoS₂ with doped transition metal elements and also characterize the electrocatalytic performance of the synthesized materials. They show that the experimental result validates the theoretical prediction showing optimal catalytic performance in the monolayer with 3Co-V.

The most interesting part of this work is that it provides useful insight into the effect of local configuration on the catalytic performance, which has not been studied much previously. However, the title of "Pt-like" is misleading. The performance, including an overpotential of 75 meV at 10 mA/cm² and a Tafel slope of 52 mV/dec, is substantially worse than that of Pt. The referee does not think the pt-like is necessary, as the major point of this work lies in advancing fundamental understanding rather than producing the best performance. Additionally, some aspects of the experimental results are not well supported.

1. The dominant population of 3Co-V in the Co-doped MoS₂ is not well supported. The authors did provide STEM images showing the existence of 3Co-V. But the STEM image may only provide information for a very small area in scale of nm². The authors should provide XPS and/or Raman to support their claim, as the different configurations are expected to give rise to difference in XPS and Raman spectra.

2. Usually an activation process is required to get stable catalytic performance of MoS₂. The catalytic performance of MoS₂ increases with cycling and then turn to be stable. The authors should comment on this issue. Did they activate the materials? Are the results provided in the manuscript collected with or without the activation process?

3. More information about the microcell measurement should be provided. Is single MoS₂ flake measured in the microcell? How did the authors measure the concentration of Co doping locally with STEM and XPS? How could the microcell measurement result be correlated to the typical three-cell measurement?

Reviewer #3 (Remarks to the Author):

In this article the authors demonstrated the possible configuration of MoS₂ by using an elemental doping method. The products were found to exhibit significantly improved (Pt-like) catalytic performance. The authors explained the performance using theoretical calculation. The study brings the HER performance of TMD materials towards the level of noble metal catalysts and is worth to further exploring.

The reviewer has the following main issues for consideration:

1. STEM/EELS Analysis: Can the authors add reference EELS spectra for each sample that are not taken from the location of the dopant? (see Fig. 2) Do the relative intensities in the HAADF images correspond to the expected dopant atoms?

A contrast intensity line scan could help elucidate this. Further, this analysis can provide evidence that substitutional doping is indeed occurring rather than some adatoms on the MoS₂ surface.

2. Dopant Uniformity and Concentration: XPS is the only method used to characterize doping

concentration (Table S3). There are flaws related to that method when used in a quantitative fashion. How confident are the authors that no residual metal ions are bonded to the substrate and adding to the signal? Calculating doping concentrations through STEM imaging would make for a much stronger case. Further, the spatial uniformity of the dopants is not clear through the high magnification STEM images provided (Fig. 2).

Please replace S12 with corresponding low-mag STEM images. This will show the doping uniformity and allow for better estimates of the doping concentration. It will also provide better estimates of relative concentrations of different defect types (i.e. single isolated TM sites vs. 3TM-Vs sites) to indeed support that the isolated sites are negligible.

3. XPS Analysis: Some of the XPS peak shifts are not self-consistent, for instance the blue-shift of Mo 3d with 3Fe-Vs doping (Fig. 2F) but red-shift of S 2p with the same doping (Fig. S8C). One expects consistent blue-shifting for Fe and Co doping because the Fermi level is shifting closer to the CB, but this is not always observed. Why/not?

Given the small magnitudes of shifts considered (<0.5 eV), perhaps plotting the peak positions of multiple samples with error bars to determine what, if any, shifts are occurring?!

Additionally, the XPS spectra of the dopant metals would help strengthen these arguments.

4. Raman/PL Analysis: How is PL impacted by these dopants? N-type doping, as many of these dopants are predicted to be, leads to an increase in the relative intensity of the trion peak. Do the authors see this?

5. Three electrode cell measurement: The 3 electrode cell measurement is unclear - more details are needed because there is no mention of the material in question. Similarly, it is surprising that the pristine MoS₂ control flake used in this measurement demonstrated "negligible" HER activity (line 220), assuming edge sites were utilized. More details should be provided for this pristine MoS₂ control to explain this result.

6. Other:

- Need more discussion of figure of merit, potentially in SI.
- Use standard crystal defect notation for substitutional replacement of Mo sites with dopants.
- The MoS₂ Raman peaks should be discussed using proper notation (E_{2g} and A_{1g}).

Questions

Have the authors considered kinetics with respect to the hydrogen adsorption free energy?

Have the authors considered or examined how pure CoS₂ or CoS₂ doped with Mo performs as an HER catalyst?

Response to Referees

Reviewer 1:

This manuscript reports on atomic-engineering of defects on the basal plane of monolayer-MoS₂, for improving HER catalytic-efficiency. The manuscript articulates the What-Why-How of its research design, with a rich set of computational and experimental data. The data interpretations are, in my opinion, at least 90% sound. Nevertheless, I am reluctant to offer the authors my favorable recommendation for the publication of the manuscript in NComms, with the following reasons:

- 1. The pillar of this work is “The intrinsic per-site activity of in-plane chalcogen atoms dominate the hydrogen evolution reaction (HER) catalytic performance of transition metal dichalcogenides (TMDs)” { the first sentence in the Abstract}. With this pillar, the structure of the work on developing the triangular-shape cobalt atom cluster with a central sulfur vacancy in MoS₂ basal plane as the best HER-catalysis is erected. I opine that this pillar is by no means rigid. Certainly, in-plane defect-engineering constitutes one approach in enhancing the performance of TMDs. However, there are other options as important as this “in-plane-defect-engineering” approach, options such as the “edge-site-engineering” option, and the “heterojunction-bilayer” option. In fact, the authors’ claim of getting the record-best Tafel slope of 57 mV/dec in their work is not valid because the “heterojunction-bilayer” option, with MoS₂/graphene, is known to give a Tafel slope of 41 mV/dec. In this context, the “heterojunction-bilayer” option should be considered as “dominating”. These prior-arts have already been reviewed by Zhu et al. {Chem Soc Rev 47, 4332 (2018)}, an article which is also cited in this manuscript.*

Response: Thanks for the reviewer’s suggestion. We agree that the edge site is also the highly active site and choosing suitable substrates can efficiently enhance the conductivity. The “edge-site engineering” and “active in-plane-site constructing” aim to manipulate the intrinsic activity of MoS₂ without the external effect (like substrates and strains). Adding substrates to form “heterojunction-bilayer”, however, is more materials engineering to improve overall performance by enhancing the conductivity of MoS₂. The work mentioned by the reviewer on MoS₂/graphene (*J. Am. Chem. Soc.*, 2011, 133,

7296) exhibits a value of 41 mV/dec in Tafel slope with the graphene as the conductive substrate. But that is not intrinsic effect. In our case, we also tried to transfer the 3CoMo₀-*V*_s sample onto the Au substrate, and non-surprisingly achieved even better performance: lowered Tafel slope (35 mV/dec) and the η_{10} value (48 mV). With the same Au substrates, the difference in intrinsic activity between 3CoMo₀-*V*_s and pristine MoS₂ are still obvious (Figure R2). In order to make our manuscript more precise and comprehensive, we added some more statements on the edge-site effect and heterojunction bi-layer in the Introduction section.

The related revisions made to Introduction part are as follows:

Page 2: “The intrinsic per-site activity of in-plane chalcogen atoms dominates the hydrogen evolution reaction (HER) catalytic performance of transition metal dichalcogenides (TMDs)” as “**The intrinsic per-site activity of in-plane chalcogen atoms plays a significant role in the hydrogen evolution reaction (HER) catalytic performance of transition metal dichalcogenides (TMDs)**”

Page 3: “The performance of pristine MoS₂ is restricted by the density of active sites^{4,7}, and the pursuit to maximize MoS₂ utility inspires researchers to explore various ways to rouse the activity of inert sulfurs in the MoS₂ basal plane. as “**While increasing the conductivity *via* forming heterojunction bi-layer with a conductive substrate can promote the overall catalytic performance^{4,6}, the performance of pristine MoS₂ is restricted by the density of active sites⁷⁻¹⁰. The pursuit to maximize MoS₂ utility inspires researchers to explore various ways to rouse the activity of inert sulfurs in the MoS₂ basal plane. For example, edge-site engineering^{9,11}, phase transformation^{12,13}, amorphization^{14,15}, and in-plane doping/vacancy modifications¹⁶⁻¹⁸ have been reported.**”

Page 4: “..., exhibiting the highest HER performance ever attained on MoS₂ materials” as “..., **exhibiting the highest intrinsic HER activity among MoS₂ materials.**”

Figure R2. a) The LSV curves, b) Tafel slopes of MoS₂ and MoS₂ with 3CoMo-Vs configuration transferred to Au substrate.

2. *In practical HER, a catalyst for the reaction must be loaded on an electrode. The authors of this manuscript take the approach of making relative large TMD layers (a scale of many microns) and decorating the unreactive basal planes with “3Co-Vs” for enhancing HER-performance. The preparation of such large flakes of TMDs typically requires CVD with adequate control of crystal growth, as shown in the manuscript. Transferring these flakes onto HER electrodes is not trivial. In comparison, many means including various solvothermal methods and CVD methods can yield nano-TMD-polygons (e.g., triangular clusters) which can be precipitated directly onto HER electrodes. When such nano-TMD-polygons merely have a few metal-atoms per edge, the number of HER-active-sites (edge-sites) per unit mass can become extremely high. In fact, a recent computational work has given quantitative details of the science and technology of this “edge-site-engineering” approach {An et al., Nano Lett. 17, 368(2017)}. In principle, the number of active sites per unit mass of TMD by synthesizing TMD-triangles (particularly for VS₂-triangles) with a size-scale not more than 2nm will be more than that by synthesizing micron-size TMD layers with around 2% of “3Co-Vs”.*

Response: *First, it is true that more edges induce better activity. However, the “edge-site engineering” and “active in-plane site constructing” are not exclusive; both mechanisms can co-exist and simultaneously promote the activity. In our manuscript, our concern is how to “wake up” the inert basal plane activity with a new mean. That’s why in our study large flakes are even better since we can exclude the edge effect. The four samples (3CoMo-Vs, 3FeMo-Vs, 1V_{Mo}, and 1Cr_{Mo}) have similar triangular side lengths but exhibit evidently different activity. So the reason lies in different local configurations in basal planes. To quantify, we can use the ratio of triangular area (nm²) and circumference (nm) as TA: TC. Our large-sized MoS₂ triangles (side length 40 μm) give TA: TC ratio of 5600. While for the small triangles with 2 nm size as mentioned by*

the reviewer, the ratio is 0.29. This is 20K difference. Hence, for large-sized MoS₂ triangles which is the case in our study, the effect of edge site is much smaller compared to the in-plane activity. That also implies that, if we apply our in-plane activation strategy to nano-TMD-polygon, we can expect way better performance.

Second, our in-plane site activity can even be better than edge sites. Let's make a comparison to the manuscript mentioned by the reviewer (*Nano Lett.*, 2017 17, 368). Their calculation shows different edge sites exhibit unequal activities (see Table R1). Among 62 different edge sites, only 7 edge sites (red boxes) show values of ΔG_H closer to 0 eV than the $3\text{Co}_{\text{Mo}}\text{-Vs}$ in our work (0.047 eV). This indicates that the activity is not only determined by the active site number but also the activity per site.

Table R1. Calculated gibbs free energy changes for hydrogen adsorption on the edges of triangle npm-MoS₂, npm-WS₂ and npm-VS₂ with metal edge terminated by S-monomer.

	ΔG_H	vertex	edge_I	edge_II	edge_III	edge_IV	edge_V	edge_VI	edge_VII	edge_VIII	active sites/ total sites	HER-site density (1/1000 amu)
MoS ₂	triangle [4]	0.28	0.48	0.55							0/12	0
	triangle [5]	-0.32	-0.10	-0.12							12/15	4.7
	triangle [6]	-0.23 ^b	-0.20	0.01	-0.33	0.00	-0.20	-0.21	0.01	-0.35	15/18	4.2
	triangle [7]	0.17	0.31	0.55	0.20	0.21	0.36	0.22			12/21	2.6
	triangle [4]	0.49	0.64	0.86							0/12	0
WS ₂	triangle [5]	-0.14	0.19	0.14							15/15	3.9
	triangle [6]	-0.20 ^b	-0.04	0.23	-0.14	0.23	-0.03	-0.03	0.23	-0.17	18/18	3.3
	triangle [7]	0.31	0.60	0.89	0.49	0.47	0.62	0.46			0/21	0
	triangle [4]		0.16	0.09	0.23						9/9	7.0
VS ₂	triangle [5]		0.16	-0.11	0.06	0.23					12/12	6.4
	triangle [6]		0.12	-0.18	-0.23	0.00	0.22				15/15	5.8
	triangle [7]		0.14	-0.16	-0.24	-0.22	0.02	0.25			15/18	4.4

^aResults on four sizes of triangles [n] with n equal to 4, 5, 6, and 7 are listed. n is the number of metal atoms per edge. ^b ΔG_H at the position of vertex* shown in Figure 1 are -0.10 and -0.20 eV for triangle [6] npm-MoS₂ and npm-WS₂, respectively.

* Table R1 is from Table 2 in *Nano Lett.*, 2017 17, 368.

3. *The scientific root of overpotential in TMD-electrochemistry is surface band-bending and location of Fermi level on the TMD-surface in the electrochemical-bath. The authors should correlate their DOS and band-structure results to overpotential-lowering, as overpotential-lowering is the value-proposition of this work.*

Response: This is a good fundamental thought. *First*, unlike bulk semiconductors, a

monolayer does not show band bending on the “surface”, since there is no bulk distinguishing from surface. Band bending in a typical heavily doped semiconductor is about 10 nm. Therefore, it may become irrelevant to calculate the band bending near the electrolyte interface. *Second*, given the fact that HER reaction rate, which is correlates with overpotential, is largely determined by the hydrogen adsorption free energy $-\Delta G_H$ (R. Parsons, *Trans. Faraday Soc.*, 1958, 54, 1053; J. K. Norskov, et al., *J. Electrochem. Soc.*, 2005, 152, J23). ΔG_H has become a well-accepted descriptor and also used for MoS₂ (B. Hinnemann, et. al. *JACS*, 2005, 127, 5308). So we used the calculated ΔG_H to explain the trend, which can also reflect corresponding overpotential trend in experiment. Note that we have further correlated the ΔG_H with the projected DOS on the reaction site and found an empirical scaling with the p center as reported in the manuscript. And the latter is one of key findings of this work.

4. The authors show the XPS spectra of Mo and claim that the spectra give peak-shifts are derived from the chemical presence of defects such as “3Co-V_S”. In reality, for analyses of semiconductors such as TMDs, XPS spectra all place their zero point of “Binding Energy” at Fermi level. Hence, the Mo XPS peak of a p-MoS₂ sample may have a Binding-Energy peak position more than 1eV lower than that of an n-MoS₂ sample. Since the “defect” concentration in this work is typically merely about 2% or so, spectral differences due to chemical-compositional changes from MoS₂ to MoS₂ having 2% defects can hardly be directly measured. The peak-shifts observed in this work must be due to the changes in Fermi level induced by defect-engineering. Indeed, the DOS computational data show that Fermi level moves towards CBM when “3Co-V_S” is introduced to MoS₂. This Fermi level movement is shown in the XPS Mo spectra.

Response: Thanks for the reviewer’s very careful inspection. We appreciate and agree. It has been reported that the dopants can induce the shift of Fermi level. As mentioned by the reviewer, the Fermi level movements in doped MoS₂ have been shown in the DOS computational data in our work, which confirms the dopant induced change in Fermi level. We have updated the description of XPS peak shifts as follows:

Page 10: “the successful incorporation of transition metal atoms into the basal plane of MoS₂, as indicated by the small shifts of Mo 3d peaks.” as “As for the the system with small doping concentration, the peak-shift is ascribed to the dopant induced Fermi level

movement³⁷.”

Reviewer 2:

Zhou et al. report the catalytic performance of monolayer MoS₂ with different local configuration of sulfur vacancies, including 3Co-Vs, 3Fe-Vs, 1V, 1Cr, and pristine MoS₂. They first performed theoretical simulation and find out preferred local configuration, which is 3Co-Vs. Then they synthesized monolayer MoS₂ with doped transition metal elements and also characterize the electrocatalytic performance of the synthesized materials. They show that the experimental result validates the theoretical prediction showing optimal catalytic performance in the monolayer with 3Co-Vs.

- 1. The most interesting part of this work is that it provides useful insight into the effect of local configuration on the catalytic performance, which has not been studied much previously. However, the title of “Pt-like” is misleading. The performance, including an overpotential of 75 mV at 10 mA/cm² and a Tafel slope of 52 mV/dec, is substantially worse than that of Pt. The referee does not think the Pt-like is necessary, as the major point of this work lies in advancing fundamental understanding rather than producing the best performance.*

Response: Thanks for this point. We removed the subjective “Pt-like” words and the related revisions are shown as follows:

Title: “Pt-like Performance of In-Plane TMD Monolayers by Configuring Local Atomic Structures” as “**Enhanced Performance of In-Plane TMD Monolayers by Configuring Local Atomic Structures**”

Page 2: “and benchmark the HER performance to Pt-like catalysts” as “**and benchmark the HER performance**”.

Page 16: “We have promoted the per-site electrochemical activity of in-plane sulfur sites of MoS₂ monolayer to a Pt-like level *via* elaborately tuning the *p* band center of sulfur (ϵ_p).” as “**We have promoted the per-site electrochemical activity of in-plane sulfur sites of MoS₂ monolayer *via* elaborately tuning the *p* band center of sulfur (ϵ_p).**”

2. *The dominant population of 3Co-Vs in the Co-doped MoS₂ is not well supported. The authors did provide STEM images showing the existence of 3Co-Vs. But the STEM image may only provide information for a very small area in scale of nm². The authors should provide XPS and/or Raman to support their claim, as the different configurations are expected to give rise to difference in XPS and Raman spectra.*

Response: Thank you for your constructive suggestion. *First*, to use the standard crystal defect notation for substitutional replacement of Mo sites with dopants, we changed the 3Co-Vs notation to 3Co_{Mo}-Vs for better clarity. To response your request for more evidence, we added the high-resolution Co 2p and Fe 2p XPS spectra with fittings to verify the contributions of 3TM_{Mo}-Vs and 1TM_{Mo} in Figure R3 (new Figure S11). We added the following discussion into the revised manuscript:

Page 11: *The single isolated TM sites account for very small ratio (<10%) compared to the 3TM_{Mo}-Vs (Figure S11, Figure S13c and Figure S14c), which corroborates that the latter are the dominating causes towards the HER activity.*

SI Page 9: *The high-resolution Co 2p and Fe 2p spectra with fitting are shown to verify the contributions of 3TM_{Mo}-Vs and 1TM_{Mo} (Figure S11). The absence of central sulfur atom in 3TM_{Mo}-Vs configurations creates more uncoordinated (TM) atoms compared to 1TM_{Mo}. As a result, the TM 2p spectra of 1TM_{Mo} should shift to higher BE position¹ compared to that of 3TM_{Mo}-Vs, which is consistent with our fitted curves. In addition, the area ratio of 1TM_{Mo} to 3TM_{Mo}-Vs is 0.1, which implies the 3TM_{Mo}-Vs configuration is the dominant populations.*

Figure R3 (new Figure S11). The fitted curves of (a) Co 2p_{3/2} of the 3Co_{Mo}-Vs and (b) Fe 2p_{3/2} of the 3Fe_{Mo}-Vs.

In addition, we offered the low magnified STEM image with thousands of atoms in the new Figure S14c to further support the dominant population of $3\text{Co}_{\text{Mo}}\text{-Vs}$. We can see that over 97 % of Co atoms were in the form of $3\text{Co}_{\text{Mo}}\text{-Vs}$. We updated the Figure S14 caption accordingly.

Figure R4 (new Figure S14c). Atomic STEM images of the $3\text{Co}_{\text{Mo}}\text{-Vs}$ configured MoS_2 samples with Co concentrations of (c) 3.8 at%. The $3\text{Co}_{\text{Mo}}\text{-Vs}$ configurations are highlighted in green circles while 1Co_{Mo} is marked by white circles.

SI Page 11: Large area STEM images of the Co-doped MoS_2 monolayer are imaged at $10\text{ nm}\times 10\text{ nm}$. As for the sample with the best HER activity (Figure S14c), the Co concentration is 3.8 at% with 97% of $3\text{Co}_{\text{Mo}}\text{-Vs}$ configurations.

3. *Usually an activation process is required to get stable catalytic performance of MoS_2 . The catalytic performance of MoS_2 increases with cycling and then turn to be stable. The authors should comment on this issue. Did they activate the materials? Are the results provided in the manuscript collected with or without the activation process?*

Response: Following a common practice, all catalysts went through an activation process of 100 CV cycles before test. Samples exhibited a minor increase in performance and then stabilized. The small difference before and after electrochemical activation process (Figure R5) indicates that no apparent active sites was created. *First*, we used graphite carbon counter electrode instead of Pt. This avoid the known “contamination issue” of Pt deposition to cathode (*J. Mater. Chem. A*, 2015, 3, 13080). *Second*, the

improved activity *via* activation usually originates from phase transformation or particle size decreasing. This cannot happen to our samples, for which dopants were atomically introduced to the MoS₂ lattice and the structure is stable. We also did not observe 2H-1T phase transition. We added the related illustrations on activation in Experimental method part.

Page 18: Before HER test, the catalysts went through an electrochemical activation process by cyclic voltammetry scanning in 0.5 M H₂SO₄ with a scan rate of 100 mV s⁻¹ under the potential range (0.2- 0.9 V vs. RHE).

Figure R5. The LSV curves of the sample 3CoMo-Vs before and after electrochemical activation.

4. *More information about the microcell measurement should be provided. Is single MoS₂ flake measured in the microcell? How did the authors measure the concentration of Co doping locally with STEM and XPS? How could the microcell measurement result be correlated to the typical three-cell measurement?*

Response: We selected the MoS₂ monolayer region with preferable large area, for the sake of as many as possible overlapping regions between MoS₂ and pre-made gold fingers for microcell measurements.

There are multiple ways to trace the Co doping concentration, including STEM and XPS. STEM and XPS quantitatively disclose the doping concentrations. Raman peak only shows positive correlation to the doping concentration. Since the doping in MoS₂ is not structurally periodic, there will be inevitable fluctuation of doping concentration all over the sample, even in single MoS₂ flake. In this sense, STEM, XPS and Raman without exception express a collective signal and average concentration of dopants, at

various scales from tens of nanometers to micrometers.

We used Raman as a pattern to differentiate doping concentration at multiple locations. From microcell samples, the Raman spectra were collected at multiple spots. Based on each Raman spectrum, we will then compare the pattern to multiple samples and select the one that shows the largest number of closest Raman patterns at multiple random positions for the XPS and STEM measurement. The XPS and STEM results are assigned to that specific spot on the microcell sample. In this way we indirectly determine the Co doping concentration locally.

Microcell measurement reveals localized electrochemical property of the material. It is a more accurate way regarding the intrinsic activity. It will not necessarily reflect the practical values as obtained in a typical three-cell measurement, where multiple geometry factors such as edges, nucleation centers and/or multilayers cannot be excluded.

Reviewer 3:

In this article the authors demonstrated the possible configuration of MoS₂ by using an elemental doping method. The products were found to exhibit significantly improved (Pt-like) catalytic performance. The authors explained the performance using theoretical calculation. The study brings the HER performance of TMD materials towards the level of noble metal catalysts and is worth to further exploring.

- 1. STEM/EELS Analysis: Can the authors add reference EELS spectra for each sample that are not taken from the location of the dopant? (see Fig. 2) Do the relative intensities in the HAADF images correspond to the expected dopant atoms? A contrast intensity line scan could help elucidate this. Further, this analysis can provide evidence that substitutional doping is indeed occurring rather than some adatoms on the MoS₂ surface.*

Response: Thanks for the reviewer's professional comment. We added the reference EELS spectra as Figure R6 (new Figure S12) that is away from the dopant site collected in the Co-doped sample as shown in Figure 2 in the manuscript. The reference spectra do not show any sharp feature of the detected element, confirming the S/N ratio is far beyond the detection limit and the observed sharp peak is not an artifact during the

collection at the dopant site. Other doped samples show similar behavior.

Figure R6 (new Figure S12). EELS spectrum in a line scan collection in Co-doped MoS₂. The spectrum was collected along a line across the dopant. A sharp peak at 770 eV appeared if we integrated the EELS signal only on the dopant position, while a clean background was found when integrated far away from the dopant (the main peaks of Mo and S are at 55 eV and 167 eV, respectively, outside the energy region that collected).

Page 11: The reference EELS spectra taken away from the dopant site (Figure S12) confirms the observed sharp peaks in the spectra of Figure 2 are not an artifact during the collection at the dopant site.

SI Page 10: The reference EELS spectra in Figure S12 have no sharp feature of detected element, confirming the S/N ratio is far beyond the detection limit and the observed sharp peak in Figure 2 is not an artifact during the collection at the dopant site. Other configured samples exhibit the similar behavior.

In order to further exclude the adatoms case, we also added the line intensity profile of the single isolated dopant site and compared them to the STEM simulation (data added as new Figure S10). In Figure R7, the simulation of the substitutional dopant configuration agrees well with the experiment. The line intensity profile of the dopant sites in the 3TM_{M0}-V_S structure is similar to the single dopant one, thus adatom can be completely ruled out.

Figure R7 (new Figure S10). Line intensity profile of the single isolated Co dopant. (a) Experimental STEM image of the single isolated Co dopant with its line intensity profile along the highlighted red line. (b, c) Simulated STEM image of the substitutional Co dopant (b) and Co adatom on top of the Mo site (c). From the intensity profile, the simulation of the substitutional dopant configuration agrees well with the experiment, thus excluding the adatom configuration.

Page 10 - 11: The line intensity profile of the single isolated dopant site and comparison to the STEM simulation (Figure S10) confirm that the TM atoms successfully doped into the lattice rather than the adatoms on the surface of MoS₂.

2. *Dopant Uniformity and Concentration: XPS is the only method used to characterize doping concentration (Table S3). There are flaws related to that method when used in a quantitative fashion. How confident are the authors that no residual metal ions are bonded to the substrate and adding to the signal? Calculating doping concentrations through STEM imaging would make for a much stronger case. Further, the spatial uniformity of the dopants is not clear through the high magnification STEM images provided (Fig. 2). Please replace S12 with corresponding low-mag STEM images. This will show the doping uniformity and allow for better estimates of the doping concentration. It will also provide better estimates of relative concentrations of different defect types (i.e. single isolated TM*

sites vs. 3TM-Vs sites) to indeed support that the isolated sites are negligible.

Response: We agree with you. We collect low-magnified STEM images as shown in Figure R8 (Fig. S13 in revised SI) and estimate the concentration of the dopants. This provides us with more direct evidence of the doping concentration and distribution. As can be seen, the TM dopants distribution are fairly uniform through the measured regime, and we can see clearly their corresponding configurations. We have made the following changes:

Figure R8 (Figure S13 in revised SI). Low-magnified STEM images of three types of doped samples: (a) 1Cr_{Mo} , $10\text{ nm} \times 10\text{ nm}$; (b) 1V_{Mo} , $12\text{ nm} \times 12\text{ nm}$; and (c) $3\text{Fe}_{\text{Mo}}\text{-Vs}$, $12\text{ nm} \times 12\text{ nm}$. The 1TM_{Mo} sites are highlighted in white circles and $3\text{TM}_{\text{Mo}}\text{-Vs}$ configurations are labeled with green circles.

SI Page 10: In order to further verify the uniformity and concentrations of dopants, the low-magnified STEM images of samples $3\text{Fe}_{\text{Mo}}\text{-Vs}$, 1V_{Mo} , and 1Cr_{Mo} are presented (Figure S13). It is clearly seen that most of the TM dopants are uniformly distributed throughout the lattice. For the $3\text{Fe}_{\text{Mo}}\text{-Vs}$, the overall Fe concentration is estimated as 3.3 %. The $3\text{Fe}_{\text{Mo}}\text{-Vs}$ configuration accounts for over 90 %. In the sample 1V_{Mo} , the overall concentration of V atoms (highlighted in white circles) is estimated as 1.4 %, and all corresponds to single V_{Mo} configuration. The sample 1Cr_{Mo} has a Cr concentration of 1.2 %.

Accordingly, we also replaced the original Figure S12 with low-magnification STEM images (new Figure S14).

Figure R9 (new Figure S14). Atomic STEM images of the $3\text{CoMo}_0\text{-Vs}$ configured MoS_2 samples with Co concentrations of (a) 1.6 %, (b) 3.1 %, (c) 3.8 at%, (d) 5.8 at% and (e) 7.0 at%. The $3\text{CoMo}_0\text{-Vs}$ configurations are highlighted in green circles while 1CoMo_0 is marked by white circles. All samples are imaged at $10\text{ nm}\times 10\text{ nm}$.

3. *XPS Analysis: Some of the XPS peak shifts are not self-consistent, for instance the blue-shift of Mo 3d with 3Fe-Vs doping (Fig. 2F) but red-shift of S 2p with the same doping (Fig. S8C). One expects consistent blue-shifting for Fe and Co doping because the Fermi level is shifting closer to the CB, but this is not always observed. Why/not? Given the small magnitudes of shifts considered ($<0.5\text{ eV}$), perhaps plotting the peak positions of multiple samples with error bars to determine what, if any, shifts are occurring?! Additionally, the XPS spectra of the dopant metals would help strengthen these arguments.*

Response: Following your question, we re-measured XPS of all samples with three times for each one. All spectra were corrected by C 1s (284.5 eV). The errors of peak shift (ΔBE) were added as shown in Figure R10.

Figure R10 (new Figure S8). The errors of ΔBE for (a) Mo 3d and (b) S 2p of the $3Co_{Mo_0-V_S}$, $3Fe_{Mo_0-V_S}$, $1V_{Mo_0}$, and $1Cr_{Mo_0}$ samples.

Figure R11 (new Figure S9). The corresponding high-resolution metal 2p spectra in the (a) $3Co_{Mo_0-V_S}$, (b) $3Fe_{Mo_0-V_S}$, (c) $1Cr_{Mo_0}$, and (d) $1V_{Mo_0}$.

Indeed, nearly all peak shifts of Mo 3d and S 2p are no more than 0.3 eV, which fall in the system error (about 0.2 eV) of XPS measurements. It is thus hard to draw a correlation (so you are right). We removed the figures of XPS peak shifts and replaced with the fine spectra of the dopants in the four samples (see Figure R9). The metal-S bonds are clearly detected in all samples. This data, together with STEM images, strongly confirm the incorporation of metal dopants as substitutional atoms. The corrections are as follows:

Page 10: However, the shifts of X-ray photoelectron spectra in both Mo 3d and S 2p are very small (below 0.3 eV, see Figure S8), likely due to the low dopant concentrations. Therefore the minor peak shift cannot justify if the dopants incorporate into the MoS_2 lattice. More evidence is provided by the high-resolution spectra of TM 2p (Figure S9), which show clearly the formation of metal-sulfur bonds in all samples and support the substitutional dopants within the MoS_2 lattice.

4. *Raman/PL Analysis: How is PL impacted by these dopants? N-type doping, as many of these dopants are predicted to be, leads to an increase in the relative intensity of the trion peak. Do the authors see this?*

Response: We fitted the PL spectra of pristine MoS₂, the 3CoMo₀-Vs, 3FeMo₀-Vs, 1CrMo₀, and 1VMo₀ to deconvolute the A exciton and trion contributions (see Figure R12). The optimal line shape for the spectral contributions is a mixed Gaussian-Lorentzian function. Overall, we find that the A exciton and trion peaks of the 3CoMo₀-Vs, 3FeMo₀-Vs, 1CrMo₀ are shifted to lower wavelength positions compared to pristine MoS₂, which is consistent with phenomenon of the n-type doped system (*Adv. Mater.*, 2016, 28, 9735). The ratio between trion and exciton (IA/IA⁻) in pristine MoS₂ is 0.17, which is significantly increased in the 3CoMo₀-Vs, 3FeMo₀-Vs, 1CrMo₀. As for the 1VMo₀, the A exciton and trion peaks are shifted to higher wavelength positions compared to pristine MoS₂ as well as the reduced value of IA/IA⁻. These results confirm the 1VMo₀ as the p-type dopant which is consistent with our DFT calculation.

Figure R12. The photoluminescence of (a) pristine MoS₂, (b) 3CoMo₀-Vs, (c) 3FeMo₀-Vs, (d) 1CrMo₀ and (e) 1VMo₀.

5. *Three electrode cell measurement: The 3 electrode cell measurement is unclear—more details are needed because there is no mention of the material in question. Similarly, it is surprising that the pristine MoS₂ control flake used in this measurement demonstrated "negligible" HER activity (line 220), assuming edge sites were utilized. More details should be provided for this pristine MoS₂ control*

to explain this result.

Response: Thanks for the reviewer’s constructive comment. The η_{10} of pristine MoS₂ shows a value of -317 mV which is the average activity among 2H MoS₂ catalysts (*ACS Nano*, 2014, 8, 10196). This value is only around 70 mV more negative than the 1CrMo sample. We also added an optical image of the pristine MoS₂ (new Figure S15), which shows the edge length of tens of micrometers. So indeed there are also edge contribution. Therefore, you are absolutely right, that our pristine MoS₂ does exhibit certain catalytic activity rather than “negligible”. For accuracy, we changed the word “negligible” to “inferior” on page 12. Thank you again.

Figure R13 (new Figure S15). Optical image of pristine MoS₂ monolayer supported on Si/SiO₂ substrate.

Page 12: “As expected, pristine MoS₂ shows a negligible HER activity with an overpotential of 317 mV at 10 mA cm⁻² (η_{10}).” as “As expected, pristine MoS₂ with an overpotential of 317 mV at 10 mA cm⁻² (η_{10}), shows an inferior HER activity than those of configured MoS₂. The optical image of pristine MoS₂ as shown in Figure S15 shows the similar edge length with that of configured MoS₂, excluding the edge effect on different activity.”

6. Need more discussion of figure of merit, potentially in SI.

Response: We have add discussion and compare with more references which are not

only the monolayer TMDs, but also bulk TMDs as well as other catalysts. A new Table S2 was created:

Page 13: In addition, a comprehensive comparison of HER performance with other TMDs and non-noble metal catalysts is made in Table S2. The performance of our MoS₂ monolayer with 3Co_{Mo}-Vs configuration exceeds all the pure TMD monolayer catalysts, and also compete with other non-noble metal catalysts.

Table S2. Comparison of HER activity of MoS₂ in this work (3Co_{Mo}-Vs) with mono/few-layered TMDs and some bulk TMDs electrocatalysts measured in 0.5 M H₂SO₄.

Sample ID	Overpotential at 10 mA cm ⁻² (η_{10} , mV vs. RHE)	ΔG_H (eV)	TOFs (s ⁻¹)	Ref.
Monolayered or few-layered TMDs				
3Co_{Mo}-Vs	-75	0.048	3-50	This work
Strained Vs-MoS ₂	-170	0.08	0.05-0.16	[2]
20 nm 2H-TaS ₂	~-200	-0.04	--	[3]
V _{Re} -ReS ₂	-147	0.016	1-10	[4]
1T-MoS ₂	<-200	~-0.15	0.2-0.5	[5]
Vs-MoS ₂	<-300	~-0.025	--	[6]
MoS ₂ on Au	<-200	--	--	[7]
2H MoS ₂ Mo edge	-201	~-0.125	3.8	[8]
Vs-MoS ₂ nanocrystals	~-150	--	--	[9]
2H MoS ₂ basal plane	-425	--	1.9×10 ⁻⁴	[8]
1T'-MoS ₂ Mo edge	-77	--	3.8	[8]
MoS ₂	~-600	--	--	[10]
Bulk TMDs and other catalysts				
Pd-MoS ₂	-89	-0.02	--	[11]
H-TaS ₂	-60	~-0.01	--	[12]
H-NbS ₂	-50	~-0.01	--	[12]
TaS ₂ 200 nm in thickness	-150	0.02	--	[3]
rGO/W _x Mo _{1-x} S ₂	-233	--	--	[13]
1T-MoSe ₂ nanosheets	-152	--	--	[14]
Co-doped MoS ₂	-156	--	--	[15]
VS ₂ crystal	-68	--	--	[16]
MoS ₂ on N-carbon nanoboxes	-165	--	--	[17]
ReS ₂ on Au foil	~-190	--	--	[18]
NiCo ₂ P _x nanowires	-104	--	0.021	[19]

7. *Use standard crystal defect notation for substitutional replacement of Mo sites with dopants.*

Response: Thank you for your careful checking We have changed the notations of 3Co-Vs, 3Fe-Vs, 1Cr and 1V to 3Co_{Mo}-Vs, 3Fe_{Mo}-Vs, 1Cr_{Mo} and 1V_{Mo}, respectively, throughout the manuscript.

8. *The MoS₂ Raman peaks should be discussed using proper notation (E_{2g}^1 and A_{1g}).*

Response: Following your suggestion, we changed to E_{2g}^1 and A_{1g} . the notation of Raman peaks of MoS₂ as follows:

Page 10: “as seen from the characteristic A_g mode at $\sim 401\text{ cm}^{-1}$ and the E_g mode at $\sim 381\text{ cm}^{-1}$ observed in pristine MoS₂ monolayer” as “as seen from the characteristic A_{1g} mode at $\sim 401\text{ cm}^{-1}$ and the E_{2g}^1 mode at $\sim 381\text{ cm}^{-1}$ observed in pristine MoS₂ monolayer”.

9. *Have the authors considered kinetics with respect to the hydrogen adsorption free energy?*

Response: In our study, we used the computational electrode model where the proton and electron are assumed to be coupled where the kinetics are not considered. But kinetics is important in the HER activity of the MoS₂ catalysts. For example, the reduced MoS₂ can have improved MoS₂ electrochemical properties (*ACS Nano*, 2015, 9, 5, 5164) indicating that the charge transfer playing a role in the overall HER performance. Theoretically, considering the charge transfer may give different active site - Mo site favors to accept electron than S site; and thus the kinetics descriptor rather than thermodynamics one should be considered (*Y. Huang, et.al. JACS*, 2015, 137, 6692). However, given the fact that HER reaction rate is largely determined by the hydrogen adsorption free energy $-\Delta G_H$ (R. Parsons, *Trans. Faraday Soc.*, 1958, 54, 1053; J. K. Norskov, et al., *J. Electrochem. Soc.*, 2005, 152, J23), ΔG_H has become a well-accepted descriptor and also used for MoS₂ (B. Hinnemann, et. al. *JACS*, 2005, 127, 5308). As a

result, we used the calculated ΔG_{H} to explain the trend and propose the descriptor since it is widely used for screening new catalysts for the first approximation.

10. Have the authors considered or examined how pure CoS₂ or CoS₂ doped with Mo performs as an HER catalyst?

Response: It is an interesting idea! Based on the literature, CoS₂ has different lattice structure from MoS₂. CoS₂ (001) plane was proposed to be most stable face for HER with the Co site as a more active site (*ACS Catal.*, 2019, 9, 456). However, both electrochemical measurement (η_{10} : 358 mV; Tafel slope: 116 mV/dec) and DFT calculations (ΔG_{H} : 0.3 eV) show that the activities are much lower than Co-doped MoS₂ in this work.

Reviewers' comments:

Reviewer #4 (Remarks to the Author):

The manuscript by Zhou et al reports a computationally-driven design of doped TMD with enhanced activity for HER. Although the authors have provided answers to previous referees (even if only partially) and comparison with literature has been made, I cannot recommend publication of the manuscript in the present form for the following reasons:

- As pointed out by Referee 1 (point 1), MoS₂ is a well-known material, studied extensively both experimentally and theoretically for HER applications. Thus, this work is more incremental than a breakthrough in this field.
- The key finding is about the correlation between the H adsorption energies and the sulfur p-band position. First, this model is very far from experimental conditions: solvent, electrolyte and double layer are neglected so in direct comparisons with experiments they may obtain the right correlation for the wrong reasons. Second, the p-band model appears as something a posteriori, since there is no a phenomenological connection between the HER catalysis and the S p-band position. In other words, the authors do not offer any physical that explains why p-bands are related to better HER catalysis. The p-band model is very famous because is an easy computable quantity and may have some physical meaning when computed for O in oxygen vacancy formation energies in oxides but here, Co is the active species and not S. Third, I am not very fond of this analysis because the absolute position of the p-band is somehow arbitrary, as it depends on the exchange correlation functional used.
- The structural model used by the authors (a 4x4 supercell) may be not sufficient for studying low concentration of large defects as the 3Co(Mo)-Vs. A convergence study is needed, such as the one done by Li and Carter in JACS 141, 10451 (2019) , where supercells as large as 5x5 are needed for converging DOS of related material WSe₂ containing single-site defects.

Reviewer #5 (Remarks to the Author):

The authors have addressed the comments raised by the original reviewers which has improved the manuscript, however I believe the following points also need to be addressed to support their arguments.

More detail on the electrochemical measurements needs to be provided – the authors state the GC RDE was covered with the catalyst – this raises the question of what was the loading? How did the catalyst stay attached to the surface during rotation and bubble evolution? Simply dropcasting a powder sample is unlikely to be reproducible.

The activation CVs need to be provided for every sample – cycling into a region of 0.90 V would certainly result in surface oxidation and probably removal of some sulphur? This needs to be checked.

The first CV for the HER for each material should be provided and compared. Activation may affect the investigated materials differently, whereby some dopants may be more unstable to such treatment. If this is the case the theoretical argument presented will not hold as the calculations will not be based on a material that is participating in the HER.

For the microcell experiments the other catalysts should be measured. Why is the experiment started at 0V, it should begin at 0.1 V like the measurements in Figure 3a.

The EIS data needs to be reconsidered, the data was recorded at 0.15 V which is a potential where the HER does not occur, therefore any discussion related to improved charge transfer between active S sites and protons is incorrect and this measurement will provide information on the conductivity of the samples. Indeed the conductivity of the different samples should be compared.

A better experiment would be to perform a galvanostatic EIS experiment at an applied current density of 10 mA cm⁻² for each of the catalysts to truly compare their performance.

Long terms constant current experiments should also be undertaken.

Overall this is interesting work but there are some key issues that detract fr

Response to Referees

Reviewer 4:

1. As pointed out by Referee 1 (point 1), MoS₂ is a well-known material, studied extensively both experimentally and theoretically for HER applications. Thus, this work is more incremental than a breakthrough in this field.

Response: We agree with you that MoS₂ has been extensively studied for catalysis, and HER is one of the important applications. The well-known obstacle that hinders the practical application of MoS₂ is its inert basal plane. Tuning the local configurations determines the activating level of basal plane. Compared to previous studies on transformed 1T' phase and amorphous MoS₂, it is more preferable to change the local configurations by creating defects in a controlled way. However, there are two main barriers to effectively changing the local configurations (see below). And the breakthrough of our work can be elaborated as follows:

i) Limitation of material database and HER catalytic activity. It is very useful while remains great challenge to introduce and observe local atomic configurations in the basal planes of MoS₂. Theoretically, local configurations with diverse combination of defects should be able to evidently tune the HER activity of MoS₂ even up to the Pt-like performance. Previous reports on the local atomic configurations in MoS₂ show only limited tuning ability and hence insufficient enhancement of HER activity (*Chem. Soc. Rev.*, 2018, 47, 4332). In our work, we achieved for the first time very stable local configurations by forming various single atom or clusters adjacent to the sulfur vacancy sites (e.g., 3Co_{Mo}-Vs, 3Fe_{Mo}-Vs). And we directly observed them by STEM images. Among various types of configurations, we established that the triangular clusters (3Co_{Mo}-Vs) has significantly increased the MoS₂ monolayer activity with a 50% reduction of the overpotential (η_{10} : 75 mV). This is the highest intrinsic HER activity among MoS₂ materials.

ii) Lack of fundamental understanding of local configurations on HER activity. Despite the numerous reports on MoS₂ for HER, there has been no direct correlation between the effect of local atomic configurations and HER activity. In the present work,

we established that the p band center of sulfur (ϵ_p) is an effective descriptor to correlate the local configurations and catalyst activities. Based on this new descriptor, one can predict the suitable local configuration in order to obtain high HER activity.

2. *The key finding is about the correlation between the H adsorption energies and the sulfur p-band position. First, this model is very far from experimental conditions: solvent, electrolyte and double layer are neglected so in direct comparisons with experiments they may obtain the right correlation for the wrong reasons. Second, the p-band model appears as something a posteriori, since there is no a phenomenological connection between the HER catalysis and the S p-band position. In other words, the authors do not offer any physical that explains why p-bands are related to better HER catalysis. The p-band model is very famous because is an easy computable quantity and may have some physical meaning when computed for O in oxygen vacancy formation energies in oxides but there, Co is the active species and not S. Third, I am not very fond of this analysis because the absolute position of the p-band is somehow arbitrary, as it depends on the exchange correlation functional used.*

Response: Thanks for the reviewer's valuable suggestions. Our calculated free energies of hydrogen adsorbed on various local configurations are the first approximation without considering experimental conditions such as solvent and electrolyte. To validate our prediction on the trend for different doping configurations, we checked the following cases as in the reviewer's concern:

We applied the implicit solvation model in vasp-sol (J. Chem. Phys., 2014, 140, 084106) and calculated the free energy for H^* on four most stable local configurations: $1Cr_{Mo}$, $1V_{Mo}$, $3Fe_{Mo}-Vs$ and $3Co_{Mo}-Vs$. The order of free energy barriers did not change under solvation correction (Figure R1). As for the double layer effect, the monolayers are used in this work instead of porous 3D materials, minimizing the double layer effect induced by porosity. In CV curves in non-Faradaic region, nearly no hysteresis loop is existed when compared to that of Pt/C (dashed box in Figure R2). The related illustrations are added as follows:

Page 6: The implicit solvation model was employed to calculate the free energy H^* . It is found that the order of free energy barriers has no change under solvation correction (Figure S4). In addition, in our experiment monolayers are used instead of porous 3D materials, which minimizes the porosity-induced double layer effect. This can be seen

from the CV curves in the non-Faradaic region (denoted by the dashed box in Figure S5) which shows nearly no hysteresis loop.

Figure R1 (new Figure S4). The calculated free energy diagram of various configurations and pristine MoS₂ with solvation effect.

Figure R2 (new Figure S5). The first CVs of the 3CoMo-Vs, 1V_{Mo}, 3FeMo-Vs, 1Cr_{Mo} and Pt/C for the HER. Compared to Pt/C, there is nearly no hysteresis loop in the CVs of the configured MoS₂ samples in the non-Faradaic range (denoted by dashed box). It implies minimized double layer contribution for the configured samples.

The correlation between the H adsorption energies and sulfur p-band is based on that the sulfur is the active site to bond with hydrogen rather than transition metals (*e.g.* Co). We proposed a bonding process in which the inert sulfur atom is first brought into an active pre-bonding state of one open valence and then binds with the H atom freely. The H-S bonding strength may then be predicted from this valence preparation energy, which in turn depends on the state of sulfur p electrons at or below the Fermi level. Therefore, the center of p band is the property that we found to correlate the hydrogen binding

strength. More importantly, some active sites by using this property as the first screening criteria and their HER activities have been realized and confirmed experimentally. The relaxed hydrogen binding with Co is shown as follows and the energy is **1.14 eV** (Figure R3), which is **much higher** than hydrogen binding with S (0.048 eV). Such high value of ΔG_H (1.14 eV) corresponds to a poor HER performance instead of high activity of $3\text{Co}_{\text{M}_0}\text{-Vs}$, demonstrating the sulfur as the active site. We added the physical explanation of our modeling in results as follows:

Page 8: “The sulfur p states around Fermi level interact with hydrogen s electron, therefore, those p states determine the H-S bonding strength.” was changed to “**We propose a bonding process in which the inert sulfur atom is first brought into an active pre-bonding state of one open valance and then binds with the H atom freely. The H-S bonding strength may then be predicted from this valance preparation energy, which in turn depends on the state of sulfur p electrons at or below Fermi level.**”

Figure R3. a) The structure of H adsorbed on Co site in the $3\text{Co}_{\text{M}_0}\text{-Vs}$ sample. b) The free energy diagram of the $3\text{Co}_{\text{M}_0}\text{-Vs}$ sample with sulfur or cobalt as active site.

The theory and simulation intend to get an insightful trend and guide the experiments to search the potential candidates. The trend is more important than absolute numbers. Hence, we revised the misleading statements as follows:

Page 4: “Given the correlation between local configurations and electronic structures, p band center of the in-plane sulfur (ϵ_p) can be tuned with good activity when ϵ_p falls into the range of -2.36 to -2.78 eV”. was changed to “**Given the correlation between local configurations and electronic structures, p band center of the in-plane sulfur (ϵ_p) can be**

tuned with desired hydrogen binding strength.”

Page 9: “Considering the window of hydrogen adsorption energy ΔE_{H^*} (-0.5 eV H^{-1} and 0.5 eV H^{-1} , light yellow rectangular area)³⁵, the acceptable HER activity can be expected determined when ϵ_p is in the range of -2.36 to -2.78 eV ” was changed to “Considering the window of hydrogen adsorption energy ΔE_{H^*} (-0.5 eV H^{-1} and 0.5 eV H^{-1} , light yellow rectangular area)³⁵, the acceptable HER activity can be expected when ϵ_p is in the range of $0.5 - 1 \text{ eV}$ higher than that of MoS_2 .”

3. *The structural model used by the authors (a 4×4 supercell) may be not sufficient for studying low concentration of large defects as the $3\text{Co}_{\text{Mo}}\text{-Vs}$. A convergence studied is needed, such as the one done by Li and Carter in JACS 141, 10451 (2019), where supercells as large as 5×5 are needed for converging DOS of related material WSe_2 containing single-site defects.*

Response: Similar with solvation effect, we checked the 5×5 supercell for all the stable configurations, the energy difference is very small (less than 5 meV as shown in Figure R4).

Figure R4 (new Figure S25). The supercell convergence test. The free energy diagram for (5×5) supercell with 1V_{Mo} , 1Cr_{Mo} , $3\text{Fe}_{\text{Mo}}\text{-Vs}$ and $3\text{Co}_{\text{Mo}}\text{-Vs}$. The free energy difference compared with (4×4) supercell is less than 5 meV .

We also checked the electronic structure calculation with 4×4 , 5×5 and 8×8 supercell with 1Co -doped MoS_2 as an example. The projected density of states on sulfur are shown in Figure R5. Therefore, we believe that the 4×4 supercell calculations are valid in both energy and electronic structures.

Figure R5 (new Figure S26). The pDOS for 1Co-doped MoS₂ with (4×4), (5×5) and (8×8) supercell. It demonstrates that the (4×4) supercell we used for the electronic structure simulations is converged and sufficient.

Page 19: The electronic structure calculation with 4×4 , 5×5 and 8×8 supercell with 1Co-doped MoS₂ as an example. The projected density of states on sulfur are shown in Figure S26. Therefore, the 4×4 supercell calculations are valid in both energy and electronic structures (Figure S25 and S26).

Reviewer 5:

1. *More detail on the electrochemical measurements needs to be provided – the authors state the GC RDE was covered with the catalyst – this raises the question of what was the loading? How did the catalyst stay attached to the surface during rotation and bubble evolution? Simply drop casting a powder sample is unlikely to be reproducible.*

Response: Thanks for these questions. The monolayer MoS₂ samples were transferred from original Si/SiO₂ substrate to GC RDE electrode rather than drop casting a powder sample. With this method, the capillary force generated during DI water evaporation strains the MoS₂ monolayers firmly to the electrode surface (*Nat. Mater.*, 2016, 15, 48; *Adv. Mater.*, 2015, 27, 4732). It has been proven that this transfer method results in very strong interface and assures the stability of catalytic performance under rotating and bubbling. The different catalyst samples have similar coverages (seen Figure R6) with an average edge length of the triangles (~30 μm). Hence, the effects of edge and number of active sites can be excluded, so that it is rationale to compare their intrinsic activities. The details were added in Materials and methods as follows:

Page 18: Polymethyl methacrylate (PMMA) methylbenzene solution was uniformly spun on the SiO₂/Si substrates on which MoS₂ monolayers were grown. After baking at 100 °C for 5 min, the PMMA film coated substrates were dipped in 5 M KOH solution. Because of increased affinity to PMMA, the monolayer MoS₂ samples together with the PMMA film were detached from the SiO₂/Si substrate as a result of the etching effect by the KOH. Then, the monolayer MoS₂/PMMA films were washed in DI water and overlaid on the GC RDE electrode. After the thorough evaporation of DI water between GC RDE electrode and monolayer MoS₂/PMMA films, the PMMA films were further dissolved by acetone, leaving only monolayer MoS₂ on the GC RDE electrode^{16,46}.

Figure R6. Optical images of (a) 3CoMo-Vs, (b) 3FeMo-Vs, (c) 1VMo, and (d) 1CrMo on

Si/SiO₂ substrate.

- The activation CVs need to be provided for every sample – cycling into a region of 0.90 V would certainly result in surface oxidation and probably removal of some sulfur? This needs to be checked. The first CV for the HER for each material should be provided and compared. Activation may affect the investigated materials differently, whereby some dopants may be more unstable to such treatment. If this is the case the theoretical argument presented will not hold as the calculations will not be based on material that is participating in the HER.*

Response: Thanks for the reviewer's comments. We checked the voltage window of activation which should be the same as that of HER measurement. We now provide the CV curves of the activation process remeasured in the new voltage window from -0.29 to 0.1 (see Figure R7). It can be seen that the samples exhibit a minor increase in the current density and then stabilize, indicating that no apparent new active sites are created. As for your question about possible sulfur oxidation, we show that even we activated the samples up to a high voltage of 0.9 V, the sulfur states in each sample do not change according to the XPS spectrum (Figure R8). And the window of 0.2-0.9 V vs RHE also corresponds to the non-Faradic region (see Figure R9). We have added a new Figure S15 and related discussions as follows:

Page 12: The stable CVs of the configured MoS₂ samples after the electrochemical activation process (black curves in Figure S15) indicate no phase change during activation.

Page 13: The electrochemical activities shown in the first CVs of the configured MoS₂ exhibit the same trend as that from LSV curves, verifying that the activation does not affect the structural stability of introduced basal configurations (Figure S15).

SI Page 11: Note that the current densities (i) of first CVs of HER with a scan rate (v) of 2 mV s⁻¹ is nearly one half or one third those of in activation CVs scanned at 100 mV s⁻¹. Theoretically, the i of CV is proportional to the square root of v , meaning that the values of i at 2 mV s⁻¹ should be one seventh that at 100 mV s⁻¹. This discrepancy can be explained by the fact that we applied rotation during the measurement of first cycle CV but not for the activation, and rotation can sharply increase the current density values of i .

Page 19: Before the HER test, the catalysts underwent an electrochemical activation process by cyclic-voltammetry scanning in 0.5 M H₂SO₄ with a scan rate of 100 mV s⁻¹ in the potential range of -0.29 – 0.1 V vs. RHE.

Figure R7 (new Figure S15). The CVs during activation process (black curves) of every 10 cycles from the 2nd to 100th cycles. The first cycles of CV for the HER with rotation at 1500 rpm of a) 3CoMo-Vs, b) 1VMo, c) 3FeMo-Vs, and d) 1CrMo. The scan rate for activation was 100 mV s⁻¹ and 2 mV s⁻¹ for the first HER CVs.

Figure R8. High-resolution S 2p of the 3CoMo-Vs with different activation voltage windows. It shows that the peak positions remain the same and no oxidation states of sulfur when the voltage window changes.

Figure R9. The CVs for the configured MoS₂ samples at the voltage window of 0.2 - 0.9 V vs RHE.

3. *For the microcell experiments the other catalysts should be measured. Why is the experiment started at 0V, it should begin at 0.1 V like the measurements in Figure 3a.*

Response: Following your advice, we have conducted new microcell measurements of the other three samples (3FeMo-Vs, 1CrMo, and 1VMo), as shown in Figure R10. All the microcell tests start from 0.1 V and we redrew the LSV curves in Figure 4c. The corrections are as follows:

Page 16: In addition, the microcell HER measurements for all the four catalysts (3CoMo-Vs, 3FeMo-Vs, 1VMo, and 1CrMo) show the same trend with that obtained from three-electrode measurements (see Figure S24 and 3a).

Figure R10 (new Figure S24). Microcell HER measurements of the 3CoMo-Vs, 3FeMo-Vs, 1VMo, and 1CrMo samples with ~1.2 % configuration concentrations.

Figure R11 (new Figure 4c). LSV curves of the samples with Co concentrations of 1.6, 3.1, 3.8, 5.8, and 7.0 at%.

4. *The EIS data needs to be reconsidered, the data was recorded at 0.15 V which is an potential where the HER does not occur, therefore any discussion related to improve charge transfer between active S sites and protons is incorrect and this measurement will provide information on the conductivity of the samples. A better experiment would be to perform a galvanostatic EIS experiment at an applied current density of 10 mA cm⁻² for each of the catalysts to truly compare their performance.*

Response: We remeasured the EIS experiments at the applied current density of 10 mA cm⁻² (Figure R12). The related corrections are as follows:

Figure R12 (new Figure 3e). Electrochemical impedance spectroscopy (EIS) Nyquist plots measured at an applied current density of 10 mA cm⁻².

Page 15: Electrochemical impedance spectroscopy with the fitted circuit model (Figure S20) shows significantly decreased charge-transfer resistances (R_{ct}) for the 3CoMo-Vs (30.3 Ω) and the 1V (47.9 Ω) samples, as compared to those of 3FeMo-Vs (91.5 Ω) and 1CrMo (100.3 Ω), indicating a facilitated charge transfer between active S sites and

protons in the electrolyte (Figure 3e).

Page 19: Electrochemical impedance spectroscopy (EIS) measurements were carried out in the same configuration at an applied current density of 10 mA cm^{-2} from 100 KHz to 0.1 Hz.

5. *Long terms constant current experiments should also be undertaken.*

Response: We conducted long term constant current experiments as shown in Figure R13. After 10 h galvanostatic test at 10 mA/cm^2 , there is no noticeable degradation (only 15 mV increase in overpotential), demonstrating an outstanding stability.

Page 15: the 3CoMo-Vs sample exhibits an outstanding long-term operation stability with minor changes in potential (Figure 3f and Figure S21), suggesting the effectiveness of the 3CoMo-Vs configuration toward HER during the whole cycling process.

Page 19: In addition, the long term constant current experiment was also conducted at 10 mA cm^{-2} .

Figure R13 (new Figure S21). Long-term HER stability of the 3CoMo-Vs sample at a constant current of 10 mA/cm^2 .

Reviewers' comments:

Reviewer #4 (Remarks to the Author):

The authors have addressed my concerns regarding the solvent (even if only explicitly) and supercell model used. Still, the analysis on the p-band of Sulphur is quite superficial. First, for "metallic" systems, as 3CoMoVs and 3FeMoVs (which are the most active sites), the p center is ill defined within DFT since it is a ground-state theory only and then energies of unoccupied states are not reliable as occupied states are. Second, the p center should be weighted with respect to the DOS, meaning that for 3CoMoVs, the band center is located probably somewhere around -4 eV, not at the arithmetic value of one half of the starting and ending point of the band. A minor comment on this is that the x and y label of Fig 1d are exchanged. I ask the authors to add caveats on the limits of their model and to perform additional analysis (e.g. on Bader Charges of the active centres before and after adsorption) searching for a more robust correlation.

Reviewer #5 (Remarks to the Author):

The authors have done an excellent job in addressing all of the comments

Response to Referees

Reviewer 4:

- 1. For "metallic" systems, as $3\text{Co}_{\text{Mo}}\text{-Vs}$ and $3\text{Fe}_{\text{Mo}}\text{-Vs}$ (which are the most active sites), the p center is ill defined within DFT since it is a ground-state theory only and then energies of unoccupied states are not reliable as occupied states are.*

Response: We thank the Reviewer for the critical comments. Our model is still new, and we are continuously working on improvement. Feedbacks from experts such as the Reviewer will be valuable to our future work. Having said that, we do believe in the validity of our model in its current version. The essence of our model is to correlate the H binding energy to the energy penalty needed for activating the S atom into an open valence state. In the cases of pristine MoS_2 , 1V_{Mo} and 1Cr_{Mo} , the S atom is in full valence. It has to lose an electron from the fully occupied p states to create an open valence. Therefore, their p center is calculated with the occupied states only. In the cases of $3\text{Co}_{\text{Mo}}\text{-Vs}$ and $3\text{Fe}_{\text{Mo}}\text{-Vs}$, the Fermi level crosses the pDOS of the sulfur p states, which means the S atom is not in full valence. Consequently, the energy penalty for creating an open valence should be significantly lower. We find that this difference can be effectively accounted for by calculating their p center using the full p band with both occupied and unoccupied states. Although DFT is known to underestimate the gap, thus could lead to unreliable energy of the virtual states as the Reviewer pointed out, it is not an issue here because there is no gap in the states we integrate. These are the reasons why our choice of p centers, though very empirical, works well for the cases studied in this work. To clarify this point, we added the following sentences in Page 9 of the manuscript.

We added a few sentences on Page 9: *“We note that our integration of the p band includes both occupied and unoccupied states for the cases of $3\text{Co}_{\text{Mo}}\text{-Vs}$ and $3\text{Fe}_{\text{Mo}}\text{-Vs}$. This is a deliberate choice which can be justified by the fact that the pDOS is continuous around the Fermi level. It also effectively raises the p band center and indicates less energy is required to activate the S atom into the pre-binding open valence state, agreeing with the fact that the S atom is not in full valence.”*

- 2. Second, the p center should be weighted with respect to the DOS, meaning that for $3\text{Co}_{\text{Mo}}\text{-Vs}$, the band center is located probably somewhere around -4 eV, not at the arithmetic value of one half of the starting and ending point of the band. A minor comment on this is that the x and y label of Fig 1d are exchanged.*

Response: Our p center is weighted with respect to DOS. It is calculated as $\varepsilon_p = \frac{\int \varepsilon n_p d\varepsilon}{\int n_p d\varepsilon}$. For the $3\text{Co}_{\text{Mo}}\text{-Vs}$, the value is -2.5 eV. In addition, we exchanged the x and y label of Figure 1d. The related correction is as follows:

Figure 1. (d) The projected density of states of sulfur in pristine MoS_2 and structured- MoS_2 ($3\text{Co}_{\text{Mo}}\text{-Vs}$, $3\text{Fe}_{\text{Mo}}\text{-Vs}$, 1V_{Mo} , and 1Cr_{Mo}) before H adsorption. The p band center is calculated as the center of the p states around Fermi level, which are showed in orange shadows in pDOSs.

3. I ask the authors to add caveats on the limits of their model and to perform additional analysis (e.g. on Bader Charges of the active centers before and after adsorption) searching for a more robust correlation.

Response: Thank you for suggestion. We performed the Bader charge analysis on the active sulfur site and plotted in Figure R1 the correlations between ΔE_{H^*} and the Bader charge change before and after the H binding, as well as the charge before binding. It looks like Bader charges can differentiate the $3\text{Co}_{\text{Mo}}\text{-Vs}$ and the $3\text{Fe}_{\text{Mo}}\text{-Vs}$ from the others by showing two separate trends. But they are not as robust as our model, where p-center puts all data points in the same trend. To calculate the p band center, the most of the S-vacancy structures, the H does not form bond with bottom S, which are not included in the model. Based on this, we added the illustration on the limits of the model to clarify the usage range.

Figure R1. The relation between the ΔE_{H^*} and the Bader charge change on the active sulfur site after hydrogen adsorption (a) and the Bader charge on the active sulfur site before the adsorption (b).

SI Page 5-6: The electrolyte effect has been neglected in our calculation. In addition, our descriptor is based on the H-S bond formation. Thus, the conditions without the formation of H-S bonding are not included.

Reviewer 5:

The authors have done an excellent job in addressing all of the comments.

Response: We appreciate the reviewer's positive feedback.

Reviewers' comments:

Reviewer #4 (Remarks to the Author):

Only because the p band center fits with their experimental data, it does not mean that it has a reliable physical meaning. At some point, they say in their response letter that " These are the reasons why our choice of p centers, though very empirical, works well for the cases studied in this work". well, that's my point all along, that something that is flawed and wrong by definition (as the energetics of unoccupied states within DFT) cannot be chosen as a descriptor even if it fits very well their data. They don't choose Bader charges (which, btw, do have physical meaning) only because they don't fit with their story. Such an arbitrary choice, knowing that the model is flawed, is not science to me and should not be published. This is even more true in a prestigious journal as Nature communications, which is a reference for scientists worldwide. The authors should publish their experimental results (very nice btw) without the calculation part elsewhere.

Response to Referees

Reviewer 4:

1. Only because the p band center fits with their experimental data, it does not mean that it has a reliable physical meaning. At some point, they say in their response letter that " These are the reasons why our choice of p centers, though very empirical, works well for the cases studied in this work". well, that's my point all along, that something that is flawed and wrong by definition (as the energetics of unoccupied states within DFT) cannot be chosen as a descriptor even if it fits very well their data. They don't choose Bader charges (which, btw, do have physical meaning) only because they don't fit with their story.

Response: We apologize for our ignorance in previous correspondence, and truly appreciate your guidance for better understanding DFT calculations. In order to clearly elucidate catalytic mechanism and establish rational descriptor, we changed to collaborate with another theoretical team and re-calculated the whole results again. The trends of the formation energies, ΔG_H , and the

effects of solvation and supercell size (thanks for the Reviewer 4 suggested reference to remind us to illustrate the supercell size issue: L. Li, E. Carter, *J. Am. Chem. Soc.*, 2019, 141, 10451) remain the same with the previous results.

However, we deeply think about the charge changes and find that it would be a rational descriptor when we take into account the total charge changes of local configurations, including both the nearest-neighbor atoms and adsorptive sulfur atom, rather than just single adsorptive atom in the previous version. More detailed arguments are as follows.

(1) MoS₂ electron delocalization affects catalytic activity. Sulfur vacancy (V_s) on the surface of MoS₂ acts as an electron donor and induces a localized gap state which make a significant contribution on catalytic activity. Below a critical carrier density, transport of these donor states is governed by nearest-neighbor hopping and variable-range hopping (VRH) at high and low temperatures, respectively (*J Am Chem Soc* **137**, 2622, 2015; *ACS Nano* **9**, 12115, 2015; *Adv Mater* **27**, 6225, 2015). Therefore, the regional charge states localized around a defect structure can accurately describe the charge transfer capacity, and make an important contribution to regulate catalytic activity than a single adsorption atom that we used in the previous version of response letter.

We added the following discussions in the manuscript:

Page 4-5: **The activation and optimization of the basal plane in TMDs have been extensively studied to achieve the stable structure and enhance their catalytic activities²⁹⁻³¹. Sulfur vacancy (V_s) on the surface of MoS₂ acts as an electron donor and induces a localized gap state. Below a critical carrier density, transport of these donor states is governed by nearest-neighbor hopping and variable-range hopping (VRH) at high and low temperatures, respectively³²⁻³⁵. It suggests that regional charge states localized around a defect structure, instead of a single adsorption atom, make an important contribution to regulating the catalytic activity. Based on the above analysis, monitoring Bader charge fluctuation induced by defects (TM substitution and S-vacancy) and H adsorption is an efficient strategy to determine the active region through DFT computational screening.**

(2) Identifying effective catalytic structure based on charge fluctuation induced by H adsorption. To identify which neighbor atoms around adsorption sulfur atom are involved, we calculated the Bader charge changes of adsorptive S site, nearest-neighbor atoms (introduced 3 TM atoms in 3TM_{Mo}/3TM_{Mo}- V_s ; introduced 1 TM atom and the nearest 2 Mo atoms in 1TM_{Mo}/

1TM_{Mo}-V_s; introduced 2TM atoms and the nearest 1 Mo atom in 2TM_{Mo}/2TM_{Mo}-V_s) and the next-neighbor atoms (S1-S7, Mo1-Mo3) as shown in Figure R1. Our calculations on 3Co_{Mo}-V_s in Figure R2 show that the nearest metal atoms and adsorptive atom should be considered as an effective catalytic structure due to relatively large charge change (Co atoms: $\sim 0.04e^-$; adsorptive S atom: $\sim 0.1e^-$). In contrast, the charge changes of next-neighbor atoms ($\sim 0.006e^-$) are much lower than those of nearest atoms, making very weak contribution to catalytic activity. This phenomenon also exists in other defect structures (Figures R3-R6), confirming the rationality of the nearest metal atoms-adsorptive S atom as an effective catalytic structure.

Figure R1 (new Figure S6). The illustration of the nearest neighbor atoms and the next-neighbor atoms in (a) 3TM_{Mo}/3TM_{Mo}-V_s and (b) 2TM_{Mo}/2TM_{Mo}-V_s, and (c) 1TM_{Mo}/1TM_{Mo}-V_s system.

Figure R2 (new Figure 1d-e). The Bader charge changes of (d) Co atoms, (e) S1 and next-neighbor S atoms, and (f) the next-neighbor Mo atoms when 3Co_{Mo}-V_s is introduced in the MoS₂.

Figure R3 (Figure S7). Bader charge analysis of $n\text{Co}$ ($n=1, 2, 3$) doped MoS_2 with or without S vacancy structures before and after H adsorption. The Bader charge changes of the nearest Co atom/Mo atoms (a, d, g, j, m) around the adsorption site S1 atom, the S1 and the next-neighbor S atoms (b, e, h, k, n) and the next-neighbor Mo atoms (c, f, i, l, o).

Figure R4 (Figure S8). Bader charge analysis of nV ($n= 1, 2, 3$) doped MoS_2 with or without S vacancy structures before and after H adsorption. The Bader charge changes of the nearest V atom/Mo atoms (a, d, g, j, m, p) around the adsorption site S1 atom, the S1 and the next-neighbor

S atoms (b, e, h, k, n, q) and the next-neighbor Mo atoms (c, f, i, l, o, r).

Figure R5 (Figure S9). Bader charge analysis of $n\text{Fe}$ ($n = 1, 2, 3$) doped MoS_2 with or without S vacancy structures before and after H adsorption. The Bader charge changes of the nearest Fe atom/Mo atoms (a, d, g, j, m, p) around the adsorption site S1 atom, the S1 and the next-neighbor

S atoms (b, e, h, k, n, q) and the next-neighbor Mo atoms (c, f, i, l, o, r).

Figure R6 (Figure S10). Bader charge analysis of $n\text{Cr}$ ($n=1, 2, 3$) doped MoS_2 with or without S vacancy structures before and after H adsorption. The Bader charge changes of the nearest Cr atom/Mo atoms (a, d, g, j, m, p) around the adsorption site S1 atom, the S1 and the next-neighbor

S atoms (b, e, h, k, n, q) and the next-neighbor Mo atoms (c, f, i, l, o, r).

We have added above data in the revised manuscript and SI as follows:

Page 7-8: It is important to reveal the underlying mechanism of enhanced catalytic activity due to the local configuration. The above analysis indicates that the defects (TM substitution and S-vacancy) and H adsorption could induce charge fluctuation of the regional structure due to electron delocalization of MoS₂. In nature, the catalytic activity principally depends on the charge transfer capacity before and after H adsorption. To identify the effective catalytic structure, we show the nearest and the next-neighbor atoms which possibly induce charge fluctuation in HER (Figure S6). On one hand, the nearest metals (nMo and doped (3-n)TM, n= 0, 1, 2) and the adsorption S1 atom have relatively large change in charge (Figure 1d-f, Figure S7-10). As a result, we consider (3-n)TM-S-nMo (TM substitutes, adsorption S atom, the nearest Mo atoms) as the first-order catalytic structure. On the other hand, the next-neighbor S and Mo atoms have a relatively small change in charge, which are considered as the second-order catalytic structure as the distance is far away from the adsorption site. As a result, it is reasonable to assume the charge regulation of the second-order catalytic structure have an extremely weak effect on that of the first-order one. The radial distributions of charge distribution are presented in Figure S7-10. Therefore, in our study, we calculate the total charge difference of adsorption S atom and nearest metals to describe the charge transfer capacity to S-H bonds.

SI Page 5-6: The illustration of the nearest atoms and the next-neighbor atoms of adsorption S1 site is shown in the Figure S6. Based on the distance from the S1 atom, the introduced transition metal atoms and the nearest Mo atoms are defined as the nearest atoms, which are indicated by red dash circles. The next-neighbor atoms are indicated by green dash circles including six S atoms and three Mo atoms.

(3) Correlating charge differences of effective structures with ΔG_H . Different defects will induce the different intrinsic character of TMDs (L. Li, Nano Lett., 2017, 17, 7962). Based on the above discussed effective catalytic structure (adsorptive sulfur atom and nearest metal atoms), we extensively calculated the charge differences of 24 TM-introduced MoS₂ and correlated them with H adsorption free energies (ΔG_H). Data are in Figure R7. A linear relation ($R^2 = 0.95$) between charge differences and corresponding ΔG_H is established for describing charge transfer capacity from catalytic structure to S-H bond. It indicates that charge difference of local structure is a rational descriptor for HER activity in TM-introduced MoS₂. Based on the correlation, we found

that the catalyst for charge difference around $0.07e^-$ possess high HER activity; This could serve as a screening criterion. In principle, this structural model describes charge balance at adsorptive site between H and surrounding environments. If we compare to the previous calculation on Bader charge of sulfur only (Figure R8), the nearest metal atoms around the adsorption sulfur atom do play an important role in catalyst.

Figure R7 (new Figure 1c). The correlation between change of Bader charge of local configuration around sulfur atoms and hydrogen adsorption free energy (ΔG_H). The dashed line is linearly fitted with $R^2=0.95$.

Figure R8. The relation between the ΔE_{H^*} and the Bader charge change on the active sulfur site after hydrogen adsorption (a) and the Bader charge on the active sulfur site before the adsorption (b) in the last revision. In this model, the effect of nearest metal atoms around the adsorption sulfur atom is not considered.

We added the related discussions in the revised manuscript and SI as follows:

Page 8: The amount of charge transfer of local configuration (atoms to induce the charge transfer includes: nearest $n\text{Mo}$, doped $(3-n)\text{TM}$, $n= 0, 1, 2$ and adsorption S1 atom) is linearly correlated

with ΔG_{H} , indicating a charge regulation effect by the local configuration on HER activity (Figure 1c). The linear correlation indicates that charge transfer capacity induced by varied local configurations are mostly delocalized in the nearest-neighbor range rather than on a single sulfur atom. We found that, a charge difference around $0.07e^-$ (which corresponds to $\Delta G_{\text{H}} = 0$ eV) should correspond to a high HER catalytic activity.

In sum, in good agreement with Referee 4, we have shown clearly that the charge difference of local effective structure is a rational descriptor of catalytic activity. So, we have rewritten the whole theory part of the manuscript. The present structural model describes the charge balance at the adsorptive site between H and surrounding atoms. We sincerely hope Referee 4 can accept our apology and relook at the revised manuscript again.

REVIEWERS' COMMENTS:

Reviewer #4 (Remarks to the Author):

The authors have change their descriptor to a change in computed effective charges, which can be only criticised on how the volume to integrate the charge around each sphere (atom) is defined, which depend on the code used. However, since the definition for each species is the same, and not absolute values but differences are compared, the analysis is unbiased and intrinsically correct for the purpose of this study.

I want to applause the huge effort that the authors have made in changing their initial computational framework, and their willingness to learn instead of getting stuck with the initial approach. I think that this is a very solid piece of work that deserves publication in its present form.

Reviewer #4 (Remarks to the Author):

The authors have changed their descriptor to a change in computed effective charges, which can be only criticized on how the volume to integrate the charge around each sphere (atom) is defined, which depend on the code used. However, since the definition for each species is the same, and not absolute values but differences are compared, the analysis is unbiased and intrinsically correct for the purpose of this study.

I want to applause the huge effort that the authors have made in changing their initial computational framework, and their willingness to learn instead of getting stuck with the initial approach. I think that this is a very solid piece of work that deserves publication in its present form.

Response: We really appreciate the Reviewer #4 to give us the chance to improve our manuscript and thank you so much for your positive comments.